# Recent Advances in Internet of Things Solutions for Early Warning Systems: A Review

**DOI:** 10.3390/s22062124

**Published:** 2022-03-09

**Authors:** Marco Esposito, Lorenzo Palma, Alberto Belli, Luisiana Sabbatini, Paola Pierleoni

**Affiliations:** Department of Information Engineering (DII), Università Politecnica delle Marche, 60131 Ancona, Italy; l.palma@univpm.it (L.P.); a.belli@univpm.it (A.B.); l.sabbatini@pm.univpm.it (L.S.); p.pierleoni@univpm.it (P.P.)

**Keywords:** Internet of Things, early warning systems, flood early warning, earthquake early warning, tsunami early warning, landslide early warning

## Abstract

Natural disasters cause enormous damage and losses every year, both economic and in terms of human lives. It is essential to develop systems to predict disasters and to generate and disseminate timely warnings. Recently, technologies such as the Internet of Things solutions have been integrated into alert systems to provide an effective method to gather environmental data and produce alerts. This work reviews the literature regarding Internet of Things solutions in the field of Early Warning for different natural disasters: floods, earthquakes, tsunamis, and landslides. The aim of the paper is to describe the adopted IoT architectures, define the constraints and the requirements of an Early Warning system, and systematically determine which are the most used solutions in the four use cases examined. This review also highlights the main gaps in literature and provides suggestions to satisfy the requirements for each use case based on the articles and solutions reviewed, particularly stressing the advantages of integrating a Fog/Edge layer in the developed IoT architectures.

## 1. Introduction

An Early Warning System (EWS) is an integrated architecture of hazard monitoring, forecasting and prediction, disaster risk assessment, communication and preparedness activities, systems, and processes that enables individuals, communities, governments, businesses, and others to take timely action to reduce disaster risks in advance of hazardous events [1]. An EWS has the following key elements: (i) risk knowledge and risk assessment, (ii) monitoring of parameters that can enhance or enable predictions and forecasts, (iii) dissemination of timely warnings, and (iv) preparedness to respond to the disaster [2,3]. The United Nations Sendai framework for disaster reduction recommends to substantially increase availability and access to multi-hazard early warning systems by 2030 [4]. In 2020, only 23 out of 195 of the UN countries had a working multi-hazard national EW system. In these countries, 93.63% of the population exposed to natural disaster-related risks was successfully protected through evacuation following the early warning [5], showing the great effectiveness of these systems. The societal impact of a national Early Warning system in terms of risk preparedness and risk mitigation are expected to be extremely relevant. A survey in California from 2016 showed that 88% of the population agreed about the importance of a national Early Warning system for earthquakes [6], and another study showed how such a system on the United States West Coast could reduce the risk of injuries by 50% by enhancing the population preparedness to the event [7,8]. From a cost–benefit standpoint, while a rigorous analysis is required for each use case and it strongly depends on the frequency of the event and the ability of the system to avoid false alarms, employing an EW system can provide great damage reduction, especially when coupled with efficient infrastructures and complementary safety measures. As such, EWSs are useful tools to protect human lives, valuable assets and the financial stability of disaster-prone regions [9]. For example, it has been estimated that a flood forecasting system can reduce up to 35% of annual damages due to floods [10]. The benefits from damage and fatalities reduction thanks to an earthquake warning system could easily repay 1 year of operation of said system [7], and the estimated benefit to cost ratio of a tsunami EWS in the Indian Ocean would be 4:1 [11]. Moreover, according to the Sendai framework, an efficient disaster risk reduction framework requires a multi-hazard approach and inclusive risk-informed decision making based on the open exchange and dissemination of disaggregated data. The use of advanced information and communication technologies could provide the means to make multi-hazard warning systems available in most countries that still do not have a national implementation, thanks to their low deployment costs, and also provide the means for smart and effective alert and information broadcasting [12]. In particular, technologies such as Internet of Things, Cloud Computing, and Artificial Intelligence can assist the monitoring, forecasting and alarm generation aspects of Early Warning (EW) by providing the tools to sense, clean, process, and analyze data coming from the environment.

The Internet of Things (IoT) consists of infrastructures interconnecting connected objects and allowing their management, data mining and the access to the data they generate [13]. It aims at connecting objects, actuators, or sensors to accomplish various tasks, such as environmental monitoring for various customized purposes [14]. A basic and generic IoT architecture includes three levels: (i) the local environment, containing smart objects or sensors that communicate with each other and interact or sense data from the environment; (ii) a transport layer that allows end-nodes from the first layer to communicate with higher layers and infrastructures; and (iii) a storage, data mining, and processing layer, usually implemented in the cloud, and possibly with systems and interfaces to let users access and visualize the data. While Wireless Sensor Networks (WSN) are an essential component in many IoT deployments (providing an interface between the local environment and the users), IoT solutions allow the coexistence of heterogeneous devices, real time applications, data analytic and data storage services, improved security [15], and energy management [16], from which WSNs can benefit. In the context of disaster management and Early Warning systems, the IoT provides the means for widespread environmental monitoring from different data sources, low latency communications and real-time data processing, which enable the generation of accurate and timely warnings in the case of disaster occurrence or forecasting.

This work presents a review on the architectures and the requirements of IoT solutions used in EW systems. In particular, the paper will first introduce a general IoT architecture and some general concepts and requirements for IoT-based EWSs, and then focus on four different natural disasters: floods, earthquakes, tsunamis, and landslides. For each of them the review will present a set of use cases that focus on or optimize some IoT-related aspects (such as radio coverage, energy consumption, fault tolerance, latency, and real-time data processing) required for the optimal operation of each specific EW context of use. The main contributions of the review article are the following: (i) it describes a generic architecture for an Early Warning system based on the IoT; (ii) it systematically determines which are the most used solutions in the four use cases examined and it highlights possible trends and gaps; and (iii) it provides suggestions for future research in this field and some recommendations to be able to satisfy the main requirements of such a system, based on the reviewed papers and the authors’ experience in this field.

Compared to other reviews, surveys, or papers containing literature overviews that focus on a particular type of disastrous event or a single category of hazards, such as [17,18,19,20,21,22,23], this paper takes into account multiple natural disasters highlighting the differences from one case to the other, in term of IoT architectures and systems requirements such as the required latency constraints. While some reviews, such as [20,21], more explicitly focused on prediction algorithms, Machine Learning prediction models and Computer Vision, or post-disaster management, the main focus of this work is on the adopted IoT solutions and their possible optimizations, including system requirements, communication protocols, data processing, and sensor network topology aspects.

The paper is structured as follows. In the next section, we explain the methodology used to compile the review paper and we introduce the Research Questions (RQs) addressed by the paper. In Section 3, a generic IoT architecture is introduced, alongside a set of requirements for an EWS. Section 4, Section 5, Section 6 and Section 7 answer the aforementioned RQs by analysing papers and up-to-date solutions for the four natural disasters that have been taken into consideration. In Section 8, we provide some recommendations about the future research on the topic of IoT solutions for Early Warning systems. Lastly, in Section 9, we discuss the main findings of the paper.

## 2. Methodology

Below we briefly analyze the methods used to compile the review and the Research Questions (RQs) taken into consideration.

### 2.1. Method

This article reviewed papers concerning IoT solutions for EW systems focus on four different natural disasters: floods, earthquakes, tsunamis, and landslides. For each of these use cases a search process of scientific articles was conducted using four different search engines: Google Scholar, Elsevier, IEEE Xplore, and MDPI. The articles collected in each search process were preliminary filtered by year of publication in order to occupy a time frame spanning from 2017 to 2022 and review more recent papers and solutions. Older key reference papers have been cited when necessary during the discussion of generic architectures, protocols, or methods. Each search process for a chosen search engine brought a great number of results, so efficient search keywords were used. The main focus of the review is on EW systems architectures and IoT-related aspects, so articles that only proposed prediction algorithms, Machine Learning models for disaster forecasting, post-disaster recovery systems, or alert dissemination have not been included among the reviewed literature, unless they also provided the implementation or proposal of an IoT architecture for EW or dealt with some of its specific aspects. The reviewed articles contain (i) proposed or developed IoT and WSN solutions, (ii) methods and simulations regarding sensor networks optimization and performance evaluation, and (iii) considerations about network topologies or other aspects related to the constraints and requirements of an EWS. Taking these aspects into consideration, specific search keywords were selected for each search process of the four use cases examined in the reviewed. For example, keywords such as “Early Warning System”, “IoT”, “Wireless Sensor Networks”, and “Alert System” were used for the research for each use case, while for a specific use case such as Tsunami Early Warning other keywords were also used, namely, “Underwater Sensor Networks”, “IoUT”, “Ocean Monitoring”, and “Sea Waves Monitoring”. Further reduction was obtained by excluding papers that did not directly deal with IoT-related topics and by crossing the specific keywords. The process of selecting research articles ends evaluating whether the selected papers could answer one or more of the Research Questions. For example, the search process of Tsunami Early Warning in the IEEE Xplore search engine brought more than 160 results, which were then reduced to 52 after the preliminary filtering by year. Subsequently, by crossing the keywords the total papers were reduced to a handful, and lastly only three papers were selected from this search engine for the review of this use case.

### 2.2. Research Questions

The paper aims to answer the following Research Questions:RQ1: What are the main constraints and strategies when developing an EWS for disaster management and forecasting, especially when it comes to IoT-related aspects?RQ2: What are the most used IoT architectures and communication protocols in EWS for different types of disasters?RQ3: How can existing EW systems be improved or optimized and what are the main gaps in literature and reviewed use cases?

The questions were selected with the objective to both give a general overview of architectures and networks in the field of EWSs, and provide possible insights or solutions during the development of such systems, starting from its prerequisites and constraints up to the overall architecture.

The goal of the reviewing process was to answer the RQs that were previously defined. For each article we highlighted the main objectives and the results, with the aim of showing both the progresses and the possible missing pieces or improvements that are needed in the more recent literature (RQ3). The reference architecture was used to analyze the papers and to easily extrapolate the main technologies used for each layer of the architecture developed in the various solutions (RQ2). Lastly, we provided some general recommendations to be able to satisfy the requirements of an EW system, based on the literature and the reviewed papers (RQ1).

## 3. IoT Architectures for EW Systems

In the following section, we introduce a simple IoT architecture that can be used to describe EW systems based on the IoT paradigm. The paper will consider this reference architecture to better describe the reviewed IoT systems in the following sections, and find possible trends. For each of the reviewed use cases, we will summarize the main features of each reviewed paper following this template, not only to highlight the key aspects of each solution, but also to show which solutions adopted Edge implementations and which did not.

### 3.1. Reference Architecture

IoT systems’ functions and peculiarities can be described starting from their architectural configuration. As for the most basic IoT solutions [24], a three-layered architecture can be used to describe a generic EW system based on the IoT. As shown in Figure 1 the common IoT architecture basically consist of a Perception layer, a Communication layer, and an Application layer [23].

While specific architectures may introduce or specify new layers and intermediate layers, such as Edge/Fog Layers, Middleware and Interface Layers, a generic and simple architecture can be considered to be one that senses data from the environment, processes it, and/or forwards it to a central server which will then use the current and previously stored data to generate alarms through different methods, such as signal processing, statistical methods, computer vision, or Artificial Intelligence (AI), specifically through the adoption of Machine Learning (ML), and Deep Learning algorithms. Below is a brief description of each layer with its main characteristics.

#### 3.1.1. Perception Layer

The perception layer has the task to sense and collect data from the environment, usually through sensors. Wireless Sensor Networks are widely used in disaster monitoring scenarios: they consist of nodes equipped with sensing units and communication units that can harvest data from the environment and then forward it towards a gateway node that interfaces and communicates with higher layers. WSNs offer benefits such as scalability, dynamic reconfiguration, reliability, small size, low cost, and low energy consumption [24]. Some aspects of a WSN development are particularly important in disaster monitoring or disaster EW scenarios, such as battery life, coverage, and fault tolerance, and they will be explained in detail in Section 3.2.

The choice of the right sensing unit can be essential in providing a timely and accurate response, and different parameters can contribute differently to a particular environmental hazard. Positioning sensors in certain zones or terrains can be particularly difficult, and while some applications monitor localized events (such as landslides), others might require deployments over large areas (such as river basins in flood EW, or the large geographical regions that can be affected by earthquakes), and this will enhance the cost of the solution and require ad hoc strategies to efficiently cover the entire area to be monitored, for example differentiating between nodes with long and short range coverage capabilities [25] or, for example, with a smart and optimized distribution of the sensors depending on the disaster probability of occurrence [26].

#### 3.1.2. Communication Layer

The communication layer transmits the data acquired and processed by the perception layer to a server, cloud service or application. This layer is responsible for routing, communication between heterogeneous networks, and reliable data transmission. There are different communication technologies that can be used to transmit data, both wireless and wired.

Wireless communication technologies in IoT solutions for EW Systems can be divided in two categories: long range technologies and short range technologies. Low Power Wide Area Network (LPWAN) technologies such as Long Range Wide Area Network (LoRaWAN), SigFox, Narrowband Internet of Things (NB-IoT) and Extended Coverage GSM IoT (EC-GSM-IoT) offer long range and can be further divided into Licensed and Unlicensed, depending on the frequency bands used. LoRaWAN and SigFox use Unlicensed Industrial, Scientific and Medical (ISM) bands, while NB-IoT and Global System for Mobile communications (GSM) use cellular networks and work in licensed spectrum. Cellular networks are widely deployed and they offer reliable services and Quality of Service, but cellular networks can be badly affected by environmental disasters [27,28], which is a critical requirement for the development of EW Systems. Among short range technologies, wireless protocols such as Bluetooth Low Energy (BLE) and Zigbee can offer low-cost solutions with very low power consumption and mesh architectures support [29]. Their main limit is the lack of support for long distance communication, unless the solution makes use of repeaters, which could enhance the costs [30]. The most common wireless communication technologies for EW systems are the following:Zigbee: Zigbee is a popular low-cost, low-energy, low-speed protocol built on existing IEEE 802.15.4 protocol and developed by ZigBee Alliance. It works on the 2.4 GHz band and it has data rates from 20 to 250 kbps. Zigbee supports star, mesh and cluster tree topologies, among which mesh connection is more flexible and reliable [31], allowing the WSN to survive node faults and node losses. It has a light weight stack compared to Wi-Fi and Bluetooth and battery life up to 5 years, but relatively short range and low data rates.Bluetooth and BLE: Bluetooth is based on the IEEE 802.15.1 standard. The ultra low-power, low-cost version of this standard is Bluetooth Low Energy. Both Bluetooth and BLE operate in the 2.4 GHz ISM band. They have data rates up to 1 Mbps and they use fragmentation to transmit longer data packets [27]. In BLE, there is a trade-off between energy consumption, latency, piconet size, and throughput, but parameters tuning allows BLE to be optimized for different IoT applications [32].6LOWPAN: IPv6 over Low-Power Wireless Personal Area Networks (6LoWPAN) is a standard defined by the Internet Engineering Task Force to send IPv6 packets over IEEE 802.15.4 or currently also over other protocols such as Bluetooth/BLE. It is widely used for sensors that need to transmit low amounts of data, and it operates on unlicensed bands. The 6LoWPAN group defined the encapsulation and compression mechanisms that allow the IPv6 packets to be carried over the wireless network to allow sensor networks to use IP instead of other proprietary technologies.Wi-Fi: Wi-Fi is a widely spread group of wireless technologies under the 802.11 standard. While faster than other IoT-specific standards such as Bluetooth, Wi-Fi devices consume more power than other devices, such as those based on BLE. Wi-Fi HaLow (802.11.ah) is a new Wi-Fi technology that operates in the spectrum below 1 GHz and is specifically designed for IoT use cases by adding low power consumption and long range, which are suitable for this kind of applications.LoRaWAN: LoRa is a physical layer technology that uses a proprietary spread spectrum technique and LoRaWAN Medium Access Control protocol is an open source protocol standardized by the LoRa Alliance that runs on top of LoRa physical layer. It works in ISM bands, that is, 868 MHz in Europe, 915 MHz in North America, and 433 MHz in Asia. LoRa’s modulation allows for great performance against interference and different data rates, from 300 bps to 50 kbps. LoRaWAN improves the received messages ratio using re-transmissions, it offers great coverage (10–40 km in rural zones and 1–5km in urban zones [33]) and low costs and long battery life for end-devices. It provides three classes of end devices for different IoT requirements, such as latency or energy consumption.EC-GSM-IoT: EC-GSM-IoT re-purposes 200 kHz narrowband carriers from GSM networks and it only requires a software update of the GSM network, without needing additional hardware. Some solutions in the reviewed literature still use GSM and General Packet Radio Service (GPRS) modules for connectivity, but Extended Coverage GSM aims to provide better performance, including better indoor coverage, large scale deployments, reduced complexity and better power consumption compared to old GSM modules and devices [34].NB-IoT: NB-IoT is a technology introduced by 3rd Generation Partnership Project that operates in licensed spectrum and reuses existing Long Term Evolution infrastructures. NB-IoT provides high coverage (20 dB stronger than traditional GSM) with a high Maximum Coupling Loss of 164 dB [34], which allows NB-IoT devices to reach underground locations (for example, for locating victims [28]). It has low energy consumption and it improves energy saving mechanisms; network procedures, protocol stack, modulation schemes, and base-band complexity are simplified to reduce the User Equipment complexity and cost. Different kinds of latency can occur during the NB-IoT communication, and latency must be kept below 10 s in real time applications [30].5G: 5G networks will provide further solutions and resources when it comes to cellular/mobile communications. Particularly, Ultra Reliable and Low Latency Communication (URLLC) aims to provide delays below 1 ms and with 99% reliability, making it particularly suitable for use cases such as Earthquake Early Warning, which is strongly characterized by the latency constraint [35].EnOcean: EnOcean works in Unlicensed bands, 868 MHz frequency in Europe and 315 MHz frequency in America. EnOcean is not capable to handle ad hoc network topologies as other wireless communication protocols and it has less features than other protocols, but its main focus is to be energy efficient [36], therefore being suitable for disaster management, especially thanks to its energy harvesting feature [27].Satellite communications: The use of satellite communications can prove effective when terrestrial communications are down or when the IoT deployment is in geographical areas that are difficult to reach with other means such as cellular communication, for example for a lack of existing infrastructures. There are some providers currently offering services that support satellite IoT, and satellites are also expected to play a relevant role in supporting 5G and IoT systems [37].

Wired technologies can also be used in WSNs and IoT systems. For example, Industrial IoT protocols such as CANOpen have also been used in the reviewed literature to connect devices and sensors that were used in the developed EW systems. Similarly, other wired system such as optic fiber communication can still prove effective for communication, for example in underwater settings. In this context, the aforementioned radio wireless protocols are often not the right choice for communications because of the different propagation scenarios, and instead Underwater Wireless Sensor Networks (UWSN) more often use acoustic communications.

Usually, wireless communications have proved to be the most efficient in disastrous events and emergencies [38], even though both wired and wireless communications are susceptible to failure. Disasters can have a large impact on infrastructures and networks facilities, for example cutting off the affected region in case of antennas, optical fiber links, or overhead cables failures [27]; as such, redundant communication channels should be considered to ensure that working communication links are always available.

#### 3.1.3. Application Layer

The application layer is at the top of the IoT layered architecture. It uses the data received from the communication layer to provide services or operations [24], possibly combining collected data with historical data, and satellite or weather forecasting data from other sources. The application layer implements algorithms to generate and propagate warnings if a disastrous event is imminent; it can provide databases to store old data and current data in real time; it can make predictions and forecasts, and so on. User interfaces can be created on top of the application layer and, in service-oriented-architectures, service management and middleware layers can be interposed between the Application Layer and the Communication layer to act as a bridge between the devices and the applications, and to ensure interoperability [39]. Cloud-based IoT platforms provide almost limitless storage and computational capabilities. There are many existing Cloud platforms that provide different services useful for IoT solutions [40]. Data analytics is an essential part of IoT EW systems, that might have to deal with large amounts of data from different sources, geographic locations and points in time that need to be processed and analyzed. Data analysis can become the bottleneck of an EW system [27], and therefore cloud platforms should be associated with modern EW systems [19]. Cloud computing also comes with problems such as latency when the amount of data to process is too big, but Fog/Edge computing can reduce the weight on the application layer. When dealing with a great number of heterogeneous devices, sensors and data sources, like in EWSs, a semantic approach can also be used to enhance queries and data processing [11].

Fog or, more generally, Edge computing can be implemented between the Communication layer and the Application layer to provide a faster response and better quality than solutions based solely on Cloud computing [24]. While Cloud services provide essential storage and processing capabilities, transmitting big amounts of data from many sensors or data sources can be costly, and processing a lot of raw data in dedicated servers will add a latency that can affect the performance of the EW system. In fog and edge computing, the data from the perception layer is first processed at the network edge (on gateways or even end devices) before transmitting it to higher layers, for example to a cloud service, so that latency and the amount of data to send to the cloud can be reduced. This can also help overcome bandwidth instability [19] (since processing data at the edge can lower the bandwidth consumption [39]) and intermittent network conditions when environmental hazards occur or during the disaster response phase [41]. Moreover, Edge Computing is also suitable for devices with limited battery life [42]. Fog nodes can also implement algorithms to make predictions based on the data collected from the perception layer [26]. It is also possible to embed ML models in Edge devices, but complexity and memory constraints could make it more challenging [39].

### 3.2. Requirements of an EW System

All IoT solutions have some constraints that need to be taken into account when deploying an IoT system. Early Warning systems need to produce well-timed warnings using data usually obtained from a WSN, which also comes with its own requirements such as limited power consumption and low power communications, high or total end-to-end reliability, and limited delays. Data transmission and processing on higher layers should also be optimized as to not add latency to the system. Therefore, the following requirements can be defined when designing an IoT solution for EW systems:Battery life: WSNs deploy sensors that need to last for a long time, especially when they are installed in locations that are hard to reach or difficult terrains that would make replacing the batteries a costly task. Energy budgets should be evaluated for each application, and data acquisitions and transmissions should be optimized to also limit power consumption in critical work conditions such as dark times operations (when solar batteries are not recharged) for sensors equipped with photo-voltaic units, or critical environmental situations that require more measurements and so on. A common energy preserving strategy is to let nodes go into sleep mode when they are not being used; however, communication protocols for WSN should be energy-efficient, minimizing overheads and re-transmissions [43].Fault tolerance and reliability: The system should be able to work even if one or more nodes are no longer available or if the network topology changes. Many factors can determine a faulty situation, such as low battery, bad coverage, a node being damaged or destroyed, etc. Since nodes or gateway mobility change the state of the network and complicate the message routing, numerous dedicated WSN routing protocols can be used to take into account these factors [43]. Protocols that support mesh network topology (Zigbee, Bluetooth) are useful because they provide flexibility for the network in case of failure of one or more nodes. Self-reorganizing algorithms and failure prediction are therefore essential to allow the EW system to keep issuing warnings [44]. Moreover, the casing or fabrication of a sensor node should be made so that bad weather conditions, floods, or hurricanes have less impact on it [27].Coverage: The geographical regions that need to be covered by an IoT solution for EW systems can be very large, and, as such, the chosen communication protocols must be able to allow long range communication between far nodes and gateways with predetermined rates, latency, packet loss, and other parameters. Some locations might also have blockage, heavy shadowing, or other issues that can compromise radio communications, and therefore a link budget evaluation is essential to understand whether or not communication links will work with the required parameters.Latency: EW systems should provide timely warnings, and as such systems should be able to transmit data quickly and the elaboration should not take time. Fog/Edge computing lowers the amount of data to be sent to higher layers, reducing the latency introduced when cleaning, analysing, and processing large amounts of data in the application layer. The choice of the right processing algorithm can also be valuable to reduce latency. Depending on the application, different time constraints could be required, and different communication protocols that are suited for EW can provide short transmission times, from the order of seconds to milliseconds.

Based on the general IoT architecture defined, and on the general requirements for IoT systems, the following sections are going to present peculiarities and solutions for each of the four use cases identified as relevant for the application of EW Systems.

## 4. Floods

Floods are one of the most dangerous environmental hazards, every year bearing enormous losses in terms of human lives and infrastructures. Flash floods are a particular concern because they happen quickly, intensely and without warning, thus requiring systems that allow to predict them and give time for evacuation and other security measures. Events that can generate floods are heavy rainfalls, thunderstorms and rapid snow melts. Hydro-geological instability and soil properties are also to be taken into account when assessing the risk of floods. Some existing and operational flood EW systems, such as the European Flood Awareness System, use rainfall detection (for example, from radars) or rain forecasts to generate alarms when the amount of rain detected has risen above a certain threshold [45]. IoT systems produce data that is immediately accessible for real-time warning applications. Prediction methods in EW IoT systems can rely on hydro-geological models or statistical and Machine Learning models that collect data in real time from WSNs, send them to a remote server for processing and then display results or generate alarms. Figure 2 illustrates a generic implementation of a flood Early Warning system based on the IoT which makes use of different types of sensors.

### 4.1. Reviewed Use Cases for Flood EW

Table 1 summarizes the articles reviewed in the context of the Flood Early Warning systems and an objective analysis of them is illustrated below.

In the scenario of flood warning systems, the solution developed by Basha et al. [25] and successfully deployed in Honduras and Massachusetts, consists of a heterogeneous WSN with different nodes that can cover large distances and avoid single points of failure. While successful, Jayashree et al. [46] pointed out that this kind of solution has the downside of needing many different kinds of nodes and dealing with heterogeneous data; therefore it is more expensive and it requires more complex computation. They propose a generic and simple EW architecture to solve this and other issues such as power consumption, resilience in absence of cellular coverage, less delay and shorter computations. The architecture consists of flow and water level sensors that send data to a server only when sensed data rises above a fixed threshold. An android app would be used for mobile users, working with Zigbee hardware that is connected to the mobiles via USB On-The-Go module for communication, so that the system works even if cellular coverage is down, but this approach would require users to be equipped with said USB On-The-Go system.

In the flood EW IoT-based architecture proposed by Sood et al. [26] the IoT layer is responsible for collecting and aggregating data from sensors and the Fog computing Layer is used to decrease latency by pre-processing the raw data from the sensors before sending it to higher layers, which is useful to ensure timely predictions and to lower the amount of data sent to the cloud. A forecasting model is also implemented in the fog nodes. The framework proposed in [26] also introduces a variable size hexagonal division of the monitored space to cover large areas while providing uniform sensing, optimized cost requirements and energy saving based on the probability of flood in each hexagonal division: by defining five probability-of-floods labels, the hexagons in which a monitored area is divided can be categorized so that only hexagons with a high probability of flood have all the sensors active, while devices in low probability areas can be put in sleep mode, minimizing energy waste.

Taking advantage of existing telecommunication infrastructures for connectivity is often one of the best and easiest solutions, as shown in the use case developed by Ibarreche et al. [47] in Colima, Mexico, which use the 3G network and the Message Queue Telemetry Transport (MQTT) protocol to send data to a remote Cloud server for data processing and storage. The system also employs drifters (mobile sensor nodes) with LoRa modules which forward data to the 3G connected nodes. The great amount of data from different data sources and the reliable communication network make the solution successful in providing timely warnings, but it comes with a high cost.

Cellular networks generally offer good coverage in most locations and future 5G deployments will further improve their performance. This is not always true, as certain locations might instead have bad coverage because of Non Line of Sight (NLoS) propagation problems or because there are not enough base stations to properly cover the entire area where the sensors are deployed. In this case, a better solution might be achieved creating ad hoc networks using protocols such as LoRaWAN, as observed by Nordin et al. in [48]. They evaluated the radio performance of both LoRaWAN and GSM in a rural environment with weak cellular coverage and with different cellular network providers in the case of GSM. They showed that 2G technology is not always suitable for this kind of application and that there is a need to migrate to newer technologies such as LTE or NB-IoT, while also proving LoRaWAN potential in this context.

Critical situations such as water levels rising over a certain threshold might require the sensors to make more measurements, with a much bigger energy consumption. For sensors equipped with photo-voltaic systems, adverse weather conditions can compromise the recharging of batteries. D. Purkovic et al. [49] took into account these factors to evaluate the energy budget of their developed solution based on EnOcean for better planning before deployment. They used a generic ultrasonic sensor which has two modes of operation: normal mode and critical mode. When the water levels rise above a certain threshold, the device enters critical mode, making more measurements and transmitting all of them. The study showed that critical mode and dark mode operations reduce the battery life to more than half its life in normal modes of operation.

Ragnoli et al. [50] also developed a solution with an heavy focus on energy consumption. Their system consists of a standard LoRa architecture, with a sensor node that communicates with a server through a gateway. The sensors used are resistance sensors, able to detect the presence of water and with a very low cost compared to other sensors, but also very simple processing since they display an on/off behavior in the presence of water. Three sensors are put at different heights in the monitored area, obtaining different thresholds and flood levels from 0 to 2. Low energy consumption is achieved using a dedicated deep sleep algorithm. Tests were made to evaluate energy consumption, especially to consider night time activity, during which the battery is not recharged by solar harvesting. Since the wake-up and data transmission phases are kept very short compared to the sleep period, it is possible to achieve good performance in terms of battery duration. While a sensor node was fully tested, the full WSN implementation was not tested, and while the system is scalable, traffic management needs to be taken into account and might not scale properly with many devices.

Protocols that support mesh networks, like Zigbee, are particularly useful for areas, such as river basins, where the central node might fail, and as such mesh implementations are recommended and have been shown to provide better reliability than star or tree architectures in terms of packet loss [54]. There should be a focus on handling nodes loss in the developed solutions to make sure the WSN keeps working under difficult operating conditions. Furquim et al. [44] propose an EW model that exhibits high tolerance to faults and node failures, especially in harsh environmental conditions. The architecture is divided into three tiers: (i) sensing nodes, (ii) a fog computing layer, and (iii) a cloud layer. Fault tolerance is obtained by providing nodes from the first two layers with self-organizing algorithms and ‘light-weight’ forecasting models that allow them to make predictions if the connection to the cloud is impossible, even though with less accuracy. If nodes from the second tier are compromised, first layer nodes take the task of aggregating the data from other nodes of the same layer. Alarms and sound systems equipped to the first layer allow to make localized alarms in case the communication system is heavily compromised. To achieve this level of fault tolerance, the model requires a large number of nodes.

Ali et al. [53] proposed a cloud based wireless sensor and actuators network to alert from floods. The proposed system is to be deployed not only to monitor water levels so that warnings can be issued, but it also includes actuators that open watchman inlets to make water flow out if needed. Water level and volume sensors identify the status of water and compare it with a threshold. Gateways are used to manage the communication between sensors and towards the Cloud, which is used to manage the entire system. An algorithm to manage the sensors and actuator was also proposed and validated through model analysis, specifying different checks for water levels: when the ‘danger’ level is reached, a warning message is generated and the watchman is activated by the gateways. While the paper focused on the WSN and overall IoT architecture, it did not propose a specific communication protocol.

Flood data is dynamic and non-linear, and as such using ML algorithms or Deep Neural Networks (DNN) to support the prediction is an often used solution, especially if previous flood and weather data from the area is available. There is a vast literature on Neural Networks and DNN used for flash flood prediction, and the approaches rely on the knowledge of previous data and flood maps and new data coming from the chosen WSN solution, for which good communication is essential, alongside the right choice of measured parameters and positioning of the network. As an example, Anbarasan et al. [55] propose a generic block diagram for flood prediction using Convolutional DNN that follows the aforementioned approach of data sensing, communication, and prediction. The developed data-processing algorithm and Convolutional Neural Networks (CNN) showed better results than systems that use Artificial Neural Networks (ANN) or Deep Neural Networks (DNN) when validated using doppler radar data.

Al Qundus et al. [51] realized in Kuwait a flood EW system that uses Machine Learning on data obtained from a sensor network to issue warnings. They deployed 24 sensors and 6 coordinators/receivers, dividing the monitored area into 6 sectors. The sensors used are water level, temperature, humidity and wind speed sensors, and the coordinator nodes also use the Google Weather API to get additional data. LoraWAN is used for communication technology. Since not many sensors were available, the number of reading from each sensor had to be raised. The data collected over a year was used to form a training set which was used to train a Support Vector Machine (SVM) model which was then implemented on each coordinator node, and validations showed a 98% accuracy for detecting flood. The thresholds for the sensors must be calculated every time a new solution is deployed in a new location (requiring a period to collect data if previous data was not available), and there is an ambiguity for data coming from overlapping endpoints, i.e., from locations that intersect, and better clustering and locations separation is needed.

As already mentioned, camera based systems and Computer Vision can also be used for flood Early Warning. Thekkil et al. [52] developed an IoT platform using simple feature extraction and image comparing algorithms that use data obtained from CMOS cameras and images from a database to produce curves that give a rank of the danger of flooding calamity. The compressed images are collected from the cameras network using Zigbee, forwarded to a GSM gateway and then sent to a processing server for analysis, with the advantage of being low-cost and economic.

### 4.2. IoT Architectures for Flood EW Systems

Referring to the layered architecture described in the Introduction, Table 2 lists the sensing units, communications technologies, and prediction methods employed, and other applications developed in each reviewed paper. The table also highlights if an Edge/Fog computing paradigm has been adopted or not in each of the reviewed use cases.

Different meteorological and hydraulic quantities can be measured for flash floods predictions, such as: water level, water velocity, amount of rainfall, temperature, and atmospheric pressure. Alongside these quantities, soil composition, topology, and soil moisture might influence floods, and weather forecast data can be used to enhance predictions. The sensors in the Perception Layer can be integrated with other sensors such as Global Positioning System (GPS) units. Smart camera systems can be used alone or alongside sensors that monitor hydraulic and meteorological quantities [56], but they require deeper data analyses and deep learning or image processing algorithms on the Application Layer (a comprehensive review on computer vision methods for flood monitoring by Arshad et al. is reported in [20]). The presence of pre-existing weather stations or hydro-geological monitoring stations can facilitate the deployment of the EW system. The correct positioning and installation of IoT devices is essential to achieve good results, and depending on the number of installed devices the solution might have drawbacks: a large number of IoT devices will increase cost and will generate redundant data; too few will decrease the effectiveness of predictions [26].

When it comes to the Communication layer, some of the reviewed solutions still rely on GSM, 3G or GPRS modules for long range communications, while new cellular solutions (such as 5G or Narrowband-IoT) could provide better coverage and less latency and an overall better performance. Ad hoc networks supported by protocols with long range such as LoRa are also a solution, and they are able to obtain high link budgets in difficult environments, but they require the installation of a gateway. Furthermore, protocols that work in ISM bands such as LoRa might experience degradation if there is a large number of devices transmitting at the same time [48,57]. Zigbee is the most used solution when it comes to short range communications, and the Zigbee support for mesh networks is considered useful to survive sensor network faults.

Lastly, while many of the reviewed solutions still make use of Cloud platforms or servers to store and process data or to run ML models, Fog/Edge layer solutions are recommended to improve the overall efficiency of the system, both in terms of latency and the amount of data to be transferred on the network. Machine Learning models can enhance the result of predictions and they can be embedded in Fog devices. More simple threshold-based alarms can also be effective to create an Edge solution and generate quicker alarms, and they are more easily implemented on local sensor nodes.

### 4.3. System Requirements and Constraints for Flood EW Systems

Important system requirements that were highlighted by the reviewed literature are: (i) coverage in remote areas and difficult terrains, which might require to discard certain IoT solutions as they might not be suitable for said environments; (ii) energy consumption, especially under certain environmental or weather conditions (such as “dark modes” of operation when using a photo-voltaic energy supply) or when certain measurement requirements are to be met; and (iii) fault tolerance and the ability of the sensor network to keep working even if some nodes fail, because of low battery or harsh weather conditions.

While the loss of nodes is taken into account in some reviewed articles and mesh solutions are proposed to avoid single points of failure, high fault tolerant systems require a large number of sensors to properly work, and most use cases do not take into account the effect of heavy rainfalls and extreme weather conditions on the communication channel, for example not planning for redundancy or fail-safe systems, as also noted in [23,58]. Link budgets evaluations are important to determine if an area can be served properly by a service (for example, with areas with heavy foliage or distant, rural areas) or ad hoc network. Services that operate in ISM bands such as LoRaWAN can provide good coverage, but they can also bring some limits due to the shared radio resource. Worst case scenarios should also be evaluated in terms of battery life, when the weather can limit solar energy harvesting or when a higher sampling frequency is required during a measurement.

## 5. Earthquakes

Earthquake EW systems can also benefit from the Internet of Things and related technologies. Sensors and sensing units can monitor vibrations and ground movement to create alarms and to alert people in certain places before the earthquake waves reach them. Even a few seconds or minutes of warning can prove essential in saving lives. The most common method to detect an earthquake is P-wave detection. When an earthquake occurs, compression P waves and transverse S waves radiate from the epicenter. Since P-waves travel faster but are have less destructive effects, if detected they can be used to forward warnings before the more destructive S waves reach a location. Figure 3 illustrates a generic implementation of an earthquake Early Warning System based on P-Wave detection.

A very simple IoT system based on this method is the one proposed by Alphonsa et al. [59], which uses accelerometers connected to micro-controllers to collect and process ground vibration measurements, and then sends the data using Zigbee to a receiver connected to a PC which is used to forward warnings to users. GSM modules can also be used to send warnings to a base transceiver station which will then alert mobile phone users.

EW systems for Earthquake monitoring are already developed in many countries, but there are still efforts to optimize sensor networks, provide reliable and low latency communications, and decrease the data processing latency, which is usually the most important factor contributing to the overall delay of an alert system for earthquakes [60]. Once an Earthquake EW message is generated by the network or processing hub, every millisecond delay to send the message corresponds to an increase of about 8 m of the radius of the area reached by the earthquake [35]. This means that decreasing latency is an essential requisite in earthquake EW and precise time constraints must be considered. Warning times and the collection of geophysical data should therefore be seconds to minutes. Since the most damage caused by an earthquake is usually localized within a certain perimeter around the epicenter, the most critical objective of an Earthquake EW system is to provide alarms within seconds to this area, while more accurate alerts to larger regions can then be sent in tens of seconds [6]. With optimized data processing algorithms and data latency reduced to 1 s, it is possible to obtain possible peak performance of 3–6 s to generate and disseminate the Earthquake EW alert [61].

### 5.1. Reviewed Use Cases for Earthquakes EW

Table 3 contains reviewed papers about Earthquake EW that focus on some important issues when designing an earthquake Early Warning system, such as reducing communication and algorithmic latency, synchronization, and the choice of WSN nodes density.

Tariq et al. [53] proposed and tested a Seismic Wave Event Detection Algorithm (SWEDA) to achieve milliseconds earthquake detection and warning. It uses inclinometer nodes for Industry 4.0 and the CANOpen communication protocol for industrial IoT. They designed and produced two types of sensor node: a flat inclinometer consisting of two accelerometers, and a cilindrical one consisting of seven sensors coupled with a 24-bit sigma-delta ADC and a programmable gain amplifier to increase the resolution. Besides P-waves, this system is also able to detect S-waves, Rayleigh waves, and Love waves, but different sensors placements are required to detect different types of waves with low computational efforts (matching positions of the sensors and direction/angular displacements of the earthquake). The data processing algorithm was optimized to avoid false alarms and reduce the computational cost of floating-point operations and calibrations, the latter being essential to avoid false alarms. The solution was tested with three different systems installed in Structural Health Monitoring (SHM) sites at Qatar University. The system was able to identify an on-site induced earthquake, with a first trigger for P-waves 11 s before a second trigger for S-waves was generated, showing good early warning capabilities.

Machine Learning models can also be implemented to predict earthquakes using data from multiple sensor sources. Fauvel et al. [62] proposed a distributed approach for Earthquake EW that uses data from GPS stations and seismometers to make predictions based on an algorithm they validated on a real-world dataset. The distributed approach is based on Edge/Fog computing and is meant to reduce the amount of data to be sent on the network by embedding a ML classifier on each sensor. The classifier produces a class output that is sent to a central server that combines the class prediction from each sensor to give a final prediction. The aggregation produced at sensor level combined with the fact that the information transmitted is unrelated to earthquake values drastically reduces the transmission effort over the network, lowering latency and making communications easier. The article showed that GPS stations and seismometers have a complementary performance (for example, for large earthquakes the accuracy is 99% for GPS, 28% for seismometers). As such, they evaluated a combined model, confirming how a multi-parametric model is the best choice in terms of accuracy.

Khedo et al. [63] simulated an on-site WSN model to predict and detect earthquakes in the island of Mauritius, and they analyzed how different WSN parameters can corrupt or enhance estimations of velocity and location of the epicenter (obtained from P waves) to evaluate the feasibility of such a system. The article focused on synchronization issues for WSNs in earthquake monitoring: it is essential that all nodes and the base station/sink are synchronized, otherwise the clock offsets between the nodes becomes a component of the wave propagation delay, thus making warnings imprecise. The simulated architecture uses Timing-sync Protocol for Sensor Networks (TPSN) to achieve synchronization between nodes. Simulation results show that while networks using the TPSN protocol for synchronization have a worse performance compared to perfectly synchronized (ideal) systems, networks with no synchronization protocols whatsoever have a much worse results in terms of velocity and epicenter localization. The paper also proves that denser networks provide better data and are more resilient to faulty nodes, and higher sampling frequencies allow better performance.

Since the efficiency of the earthquake detection relies on the network’s density, this could translate into a very high cost for the Earthquake EW solution. Micro-ElectroMechanical System (MEMS) can provide a low cost alternative to traditional seismographs or seismometers and they can be used to build denser networks with a smaller investment. Fu et al. [64] proposed a MEMS accelerometers network that can meet the same performance as a classic dense Earthquake EW network. Each low-cost MEMS Seismograph consists of: the MEMS accelerometer module, TCP/IP module, Power over Ethernet module, and an optional local storage module. Synchronization is achieved using a simplified Network Time Protocol (NTP). The standalone TCP/IP module is able to handle the data communication to the server via Ethernet by itself, so that the sensing unit can be controlled by a simple MCU unit, since it does not have to handle the communication routines. A low-cost seismic sensor array with 10 sensors connected to 3G/4G modems was built for field testing. The records obtained by the low cost sensors have good consistency with the data obtained by the standard seismographs and that they can obtain clear seismic phases to trigger earthquake detection for early warnings, and even earthquakes with smaller magnitude (M 3.1 to M 3.6) were identified within a 20 km range from the sensors. Bigger earthquakes (M 4.7) can be detected at up to 200 km from the sensors.

Peng et al. [65] tested the performance of a dense Earthquake EW network of 170 sensors that they were able to deploy in the Sichuan–Yunnan border region, China. The sensor design is better specified in [66] and it is a low-cost, low-energy MEMS accelerometer sensor with the performance of a Class B accelerometer (for comparison, MEMS sensors are usually Class C sensors, such as the unit used in the previously reviewed solution [64]). Each unit also provides data suitable for P-wave detection, early warning parameters computing, and a low-latency data packet transmission. For each station, a 3G/4G router was used to transmit the ground-motion data recorded by the network to a processing server. The records obtained by the sensors were compared with the ones of classic earthquake stations, showing a good consistency, even for sensors at more than 150 km from the epicenter, thus proving that this low-cost solution can be used for dense Earthquake EW system and allows drastic reduction of the investment required to cover the region of interest. The system is also able to produce ‘shake-maps’, which are also useful for post-disaster response. When the seismic network in the earthquake source region is sparse, there are large system biases when computing the shake-map. Moreover, usually more than ten minutes are needed for generating a shake map, while the deployed high density network computed shake-maps with good accuracy and almost in real time.

MEMS sensors are installed in most smartphone devices, so the information from a large number of devices can be used to create a widespread seismic network. This is called mobile crowd-sensing paradigm, or MCS. An MCS architecture is the one proposed by Zambrano et al. [68] which consists of three layers: layer 1 collects data from the low-cost smartphones network, including the users’ position using the devices’ GPS units, and detects seismic peaks, achieving synchronization between devices with different time references using the NTP; layer 2 determines if there is an actual seismic event to notify and it ensures the global reliability of the system; layer 3 is the control center, and it communicates with emergency management centres. The high heterogeneity of the sensor network also requires standardization of the sensor data and its collection, which is achieved at the second layer by the Sensor Web Enablement framework and specifically by its Sensor Observation Service component. The architecture also takes into account battery saving and memory consumption for the user device, reaching good battery performances thanks to the use of the lightweight MQTT protocol. The seismic detection was also validated using real data, showing that the solution anticipates the seismic peak by 12 s.

The MyShake Platform is an operational framework that can provide Earthquake EW system using smartphones developed by Allen et al. [69] and that has been working as a phone application since 2016. Alerts can be generated by either detecting p-waves using the sensors embedded in the user’s smartphones or mining data from regional seismic networks (for example, it currently receives data from the ShakeAlert system in the US). While the smartphones approach makes the deployment of the network very easy, characterizing the parameters of the earthquake (location, magnitude, and origin time) is dependent on the number and geographic distribution of phones with the MyShake app around the event. An ANN embedded in the smartphone app is used to distinguish between earthquake-like ground motions and everyday motions. When an earthquake event is triggered on the phones, the MyShake server looks for space–time clusters of triggers to confirm that an earthquake is underway. While the app can provide earthquake detection, location, magnitude estimation, and shake maps, current effort is on improving its early warning capabilities. A simulation platform was created to test the early warning capabilities of the system. The simulation showed that an issue can be alerted 15.6 s before a densely populated area is reached by the earthquake, but there are 2.8 s of delay to forward the alert from the MyShake server to the cellphones, and it is not clear how this delay would scale with a large number of devices in a real scenario. The predicted magnitude was lower than the observed one. The new version of the app MyShake2.0 will issue and receive earthquake early warning alerts and improve the overall characteristics of the app.

Klapez et al. [67] began the development of a low-cost, low-power and cloud-based earthquake alert system called Earthcloud which uses geophones instead of MEMS accelerometers, claiming they offer less noise at lower sampling frequency. When variations in ground motion are detected, sensors send data to a server and also produce a first alarm if the motion is strong enough. Specifically, the data is encapsulated into an MQTT packet and forwarded to the Amazon Web Services (AWS) IoT Core which processes and then sends them to Amazon Kinesis. Amazon Kinesis is used for pre-processing but also to forward data in real-time to both connected devices (which would receive a second confirmation alarm or a first alarm if the sensors initially did not trigger any alarm by themselves) and Amazon S3, a data storage service. The developed solution was tested using a three-unit network in the city of Modena, Italy, and only one out of the three sensors failed to detect an earthquake event, while the others were successful.

D’Errico et al. [35] propose the architecture of an EW system that uses the 5G network to achieve low latency in transmitting data from a Structural Health Monitoring WSN to a server for data processing. By providing certain services with dedicated resources (“network slicing”), the 5G New Radio is able to support URLLC services that allow devices to send data with delays below 1 ms and with high reliability. Software Defined Networking, Network Function Virtualization and Mobile Edge Computing further enhance performance by being able to allocate resources dynamically to each service and deploy computational resources nearer to the end users.

Hung et al. [70] also developed an Earthquake EW system based on SHM, but with a focus on energy efficiency, which remains a key constraint in Earthquake EW, too. A sentry node was integrated into the gateway unit of a WSN that used the IEEE 802.15.4 standard. The sentry node is composed of a P-wave detector, a Wake-on Radio (WoR) transmitter, and a data sink radio. The WoR devices mounted on each node monitor the channel for wake-up signals, which are used to reduce the energy consumption in Rendezvous phases, i.e., when a receiver must wake-up when a transmitting node on a higher layer initiates a communication. The use of WoR devices allows the main processor of each node to go completely into sleep when no transmission is required. The proposed architecture consists of: (i) a sensor gateway based on 3G/Wi-Fi access point, (ii) a p-wave detector connected to the gateway, and (iii) wireless sensor nodes with WoR-Receivers organized hierarchically. The P-wave detector triggers the more reliable and accurate WSN system through the WoR devices when a P-wave is detected. The study shows a 229 ms delay to the wake-up command, but the proposed scheme has a shorter wake-up delay and is more energy efficient (with power consumption 350 μA) than IEEE 802.15.4 (which has power consumption in beacon mode from 1.43 mA to 2.51 mA, depending on parameters set).

### 5.2. IoT Architectures for Earthquake EW Systems

Table 4 summarizes the technologies used on each layer of the reference architecture for the Earthquake EW use cases reviewed.

Most of the reviewed systems monitor ground motion to detect earthquakes, and as such accelerometers are the most widely used sensors. Other methods like using GPS displacement data can be also useful for real-time monitoring and early warning [6]. Structural Health Monitoring systems that use MEMS sensors to measure building motion can also be used as data sources for the evaluation of ground motion and its effects on buildings at an affordable price [71,72]. MEMS sensors are widely spread and cheap solutions and, while the quality of data they provide is less accurate than dedicated seismic devices, they can be used to collect large amounts of data from different sources and they allow to make predictions with the right data processing and conditioning. Since denser networks proved to be more effective in predicting earthquakes, using MEMS sensors can be a valuable asset to be able to install more sensors while keeping low costs.

The reviewed solutions use both short range protocols such as Wi-Fi and long range cellular protocols, while some solutions also use wired local area communication solutions (Ethernet), even though wireless communications have proven to be the most effective in case of emergencies [38]. With the advent of 5G, communications could be further improved, and URLLC services satisfy two of the main requirements of Earthquake EW, latency and data reliability. Earthquakes can be particularly damaging to network infrastructures, cutting off entire areas from communication networks and making it difficult to send data from wireless sensor networks, that could also be damaged by the earthquake itself. Moreover, network traffic usually rises after an earthquake, requiring operators to employ schemes to handle congestion and high traffic. Satellite communication provides wide coverage and it is not affected by ground disasters, so they can guarantee good communications in cases of paralysis of primary communication networks [28].

When it comes to detection algorithms and data processing, P-Wave detection is usually the easiest method to be implemented. Moreover, it allows to run earthquake detection algorithms on memory constrained devices to create on-site warning systems and to implement faster Edge solutions. Fast algorithms should be used to avoid adding data processing latency to the system.

On-site methods are useful for detection of earthquakes near a seismogenic zone [66]. In particular, STA/LTA algorithms are especially suited for real-time monitoring and early warning, especially when the devices have limited memory or processing power [73]. Other methods such as Machine Learning can also be effectively used. Compared to the other use cases reviewed in this article, the papers for Earthquake EW made a larger use of Edge solutions, which are essential to reduce the latency of the system to the minimum, for example allowing the nodes to detect earthquakes and send warnings without processing the ground motion data at a central server or Cloud, where the data can still be used for more accurate estimations later without reducing the overall speed of the alert. Moreover, employing an Edge Computing approach, particularly a distributed one like the system proposed in [62], could be useful to avoid issues deriving from network outages during large earthquakes by distributing the earthquake detection capabilities over various devices disseminated on the territory.

### 5.3. Systems Requirements and Constraints for Earthquake EW Systems

Because of the moderate seismicity across Europe, the main focus and requirement for European Earthquake EW has been on speed rather than source characterization, though research on this aspect is increasing [6]. Meantime, low latency (in terms of both data processing, chosen prediction algorithms, and data transmission) should always be a focus as to provide warnings before the main earthquake event occurs. In particular, data latency is often the most important factor contributing to the delay of an Earthquake EW system [60], and as such the packetization format, data serialization and compression methods, the structure of the seismic network and datalogger, etc., should be optimized.

An Earthquake EW system generally has limitations in terms of accuracy of prediction and detection, which may lead to both false alarms and missed warnings. An alerting threshold lower than the damaging threshold (or multiple thresholds) will produce more false alarms, but it will minimize the missed alerts and possible damage to the user [74].

Other aspects that came to light from the reviewed papers were: (i) the essential role played by synchronization and the subsequent need for synchronization protocols in the deployed WSNs; (ii) the need for fault tolerant sensor networks; (iii) the correlation between node density and performance, both for quality of prediction and resilience, since WSNs show less data corruption after some nodes fail [63]. Many efforts in the reviewed literature were focused on the design of low-cost sensor networks that could allow a denser deployment of seismic networks in this regard. Density also plays a factor in the overall latency of the system [60]. Redundancy for both devices and communication channels should be considered while retaining low costs [67], and improved battery life should also be an objective. Since latency is a very essential requirement, Fog/Edge computing can be exploited to reduce data transmissions and computational times at the main server, for example embedding Machine Learning models or prediction algorithms in sensing devices or gateways to make a first prediction on a lower layer.

## 6. Tsunamis

Tsunamis are large waves in a body of water generated by different events, such as earthquakes, volcanic eruptions, landslides impacting the water, or underwater explosions. Tsunami prediction involves seismic event detection and sea water levels and wind-waves measurements. Other methods can also be used in a Tsunami EW system, such as hydro-acoustic waves measurement, pressure measurements or camera based methods. In tsunamis generated by seismic events, the initial warning could be given by earthquake measurements, to locate the epicenter and make a prediction on the tsunami time of arrival, then other measurements can take place, such as hydro-acoustic waves detection [75]: earthquakes or, more generally, movements in the sea bottom, generate hydro-acoustic waves that propagate at the speed of the sound in water (1500 m/s) and, since their propagation speed is much faster than that of the surface waves, they can be used to predict tsunamis deploying underwater acoustic sensors. Similarly, underwater pressure sensors have also been widely used to detect tsunami waves [76].

Generally, Underwater Wireless Sensor Networks can be used to collect and record data from large geographical settings and forward it to a central hub for data processing and generation of an EW. UWSNs are used for many applications, and Internet of Underwater Things (IoUT) is a paradigm that allows detection and prediction of events that could lead to disasters. A UWSN usually consists of the following components: (i) underwater static or mobile nodes equipped with sensing units and acoustic modems; and (ii) sink nodes (on autonomous surface vehicles, buoys or ships) equipped with acoustic and radio modems [77], used to gather data from the underwater sensors and forward it to a remote server or monitoring center, usually through an IP network [78]. The use of acoustic modems instead of radio units is usually the better choice because of the propagation conditions in an underwater environment. Figure 4 illustrates a generic implementation of an underwater Tsunami Early Warning System based on the aforementioned basic UWSN architecture.

### 6.1. Reviewed Use Cases for Tsunami EW

Table 5 contains reviewed articles on Tsunami EW systems, including both proposed or tested solutions and methods to evaluate and optimize sensors deployments, especially for the more critical underwater solutions. Non-UWSN systems and designs are also reviewed.

Freitag et al. [79] designed and tested the performance of an acoustic underwater system to develop a near-field Tsunami EW in the Mentawai Basin, between West Sumatra and Indonesia. In this application the distance between sensors (pressure sensors) is large (tens of kilometers) and the distance to shore even larger, so a lower frequency is used for communication. The system was deployed close to the bottom at different heights and with variable receiver ranges. The communication link demonstrated a high reliability with a 350 bps data rate, showing that it is feasible for a tsunami alert system. Higher data rates could be achieved but they would have a lower reliability (which could require re-transmissions of sent packets). The latency is of about 1 s, so the time needed for data to be available to the user or to be ready to be transmitted to another node (in multi-hop configurations) is the duration of the packet plus 1 s, which is a good result for early warning in this context.

The study by Meza et al. [80] presents a method to determine an optimal array configuration of offshore tsunami sensors for near-field tsunami early warning. The methodology was tested in Northern Chile. The main issue with near-field tsunamis is that there is a short time interval between the tsunami generation and arrival, so it might be difficult to determine the tsunami source in time. Sensor cost might also be an issue so the study wants to find the minimal number of sensors for an EW system to get optimal performance. Factors that influence a prediction include: position of the sensor compared to the main wave energy beam (parallel or orthogonal to the wavefront); time of arrival of the pressure wave to the sensor (sensors closer to the event have a smaller observation time and therefore less data available for predictions); depth of the sensor (must be high enough as to ignore non-linear effects); position of the sensors relative to each other (the network granularity is chosen so that it is higher than the tsunami’s wavelength). Results showed that a three-sensor configuration can provide accurate estimations of the tsunami arrival time and peak amplitudes for the first wave.

Already existing national underwater stations or sensor networks (both wireless and wired) can be used to collect data (for example, about ocean bottom motion) and use it for predictions and to issue warnings. S-net is a dense cabled underwater observation network deployed along the Japan trench. It consists of 150 observation units, including both seismometers and bottom pressure sensors, connected by cables at 30 km intervals. All the data arrive to land in real time using optic fiber communications [87]. Inoeu et al. [81] developed a method for near-real time tsunami forecasts using the S-Net sensor network data, observing the wave-forms obtained from the pressure sensors. The method is able to classify the sensors of the S-Net in terms of distance from the uplift region of the earthquake by observing the registered pressure wave-forms, and, by observing which sensors are near, outside or far from said region, it is able to compute the area of the uplift region. This technique can be used to estimate both the source region of the tsunami and the earthquake magnitude. Since earthquake waves and hydro-acoustic waves are also detected by the network, filtering is required to obtain only the pressure data. The considered time-window for computing the estimation is of about 500 s after the initial earthquake, and only 1 min is required to classify the sensors, and about 10 min to compute the source area of the tsunami after the earthquake, without the need of complex simulations. The technique was successfully validated on two past large earthquakes in that region.

Jain et al. [82] propose a framework and a prediction algorithm for Tsunami Early Warning in the Pacific Ocean that measures different kinds of parameters that are effected by tsunamis and monitors marine wildlife behaviour. Since changes in the magnetic field are generated by a tsunami wave, magnetic sensors can be used to detect said changes as precursors of tsunami waves; the wave itself exhibits particular waveform patterns that can be measured by a Tide Gauge; count sensors and motion sensors are used to monitor wildlife which is also affected by changes in magnetic fields from tsunamis. This data would be sensed by a WSN on the sea bed, then transmitted to a base station, and then processed at the main server. Predictions are computed by a Machine Learning Algorithm (logistic model trees) and the study had shown that the most relevant features for the prediction are tide levels changes and migration pattern changes.

Gardner-Stephen et al. [83] propose a model for Tsunami EW to improve some aspects of existing warning systems, especially for what it concerns the cost of the solution (which can be prohibitive for certain countries) and the communication in compromised situations. The proposed solution includes both an EW system (based on autonomous underwater vehicles), a platform that acts as a ‘Warning Decision Support System’ and is able to work offline (when disastrous events cut off standard internet communications), and an alert distribution system. The EW system makes use of low cost vehicles (which were produced and designed by one of the authors) that are characterized by low mobility, a mostly passive operational mode with very low power consumption (1 Watt, allowing weeks of operation before recharge) and the possibility to sense tsunamis remotely using acoustic detection (hydrophones). The vehicles network can also cooperate with traditional fixed tsunami sensors and they are equipped with a GPS unit and satellite connectivity. Satellite communications are also used in the Alert Distribution and Support System for the aforementioned possibility of standard IP based connections failing during the disastrous events.

Underwater sensor networks are not the only available solution for Tsunami EW. Adi et al. [84] designed an IoT device for wave height monitoring using an ultrasonic sensor. The device is supposed to be installed on the shore line, directly above water. When the distance between the sensor and the water raises above a certain threshold, the GSM module mounted on the IoT devices sends an SMS with a warning message before the water level becomes dangerously high. A sound indicator is also installed on the device to alert nearby people. The prototype was successfully tested but the ultrasonic sensor cannot read any water level data if put at a distance higher than 3.19 m from seawater, and as such specific data about seawater conditions are needed before installation.

Kato et al. [85] designed a buoys satellite system for ocean monitoring and tsunami early warning. The first deployed system consisted of a buoy equipped with a Global Navigation Satellite System (GNSS) antenna able to transmit data to the coast before the wave reaches it. The received data (sea level, buoys position) allows users to visualize the motion of the sea surface at the buoys in real time. The development of this solution began in 1997, with subsequent deployments in 2001–2003, 2004–2006, and 2011. These previous solutions showed that buoys placed 20 km from the coast did not provide enough time to warn the coast and evacuate it. They also provided data to address some critical issues about buoys design, with the goal of each buoy surviving for at least 10 years. Since the required buoy distances could be of above 100 km, the Precise Point Positioning algorithm is proposed to know the position of the buoys alongside satellite communications. Field experiments showed accuracy of a few centimeters in buoys positioning. One early experiment showed data gaps, possibly due to the tilting of the buoy degrading the communication, so a mechanism to stabilize the antenna was used in subsequent ones. The paper also proposes the use of buoys to make ocean bottom motion measurements, which provides less accuracy than solutions such as S-Net but better resolution. The experiment used three transponders placed on the bottom of the ocean and communicating with the buoys with an acoustic link. Ocean crust motion measurement can be obtained by estimating the transponders position.

Darmawan et al. [86] designed an IoT device to generate early warnings when anomalies in wave behavior are detected, using Fuzzy Logic for accurate predictions. The sensor node consists of a gyroscope, an accelerometer, and a LoRa module for data transmission. The sensor floats in the sea and senses and processes data to obtain information on wave height and wave speed. These two parameters are then used to classify the wave using the Fuzzy Logic algorithm embedded in the sensor unit. Sensed data is then sent to a server using the LoRa module. The classification algorithm provides both a danger label (safe, standby, dangerous) and a danger level (a value from 0 to 100) by computing height and speed of the waves. The Fuzzy Logic algorithm bears a 98% to 100% accuracy in testing phase. The LoRa communication was also tested, with 4.6279 s of delay and minimal error rate.

### 6.2. Iot Architectures for Tsunami EW Systems

Table 6 lists the main technologies used in Tsunami EW reviewed literature for each layer of the reference IoT architecture.

Detection of wave heights and anomalies on the sea surface can also be used for early detection of tsunami waves. Wave measurements can be obtained from offshore buoys, but while they are very accurate, they can also be very costly. Low-cost, low-power MEMS sensors, such as accelerometers and gyroscopes, can also be used to obtain wave characteristics such as height, period, direction and speed by deploying them on floating buoys.

The use of sensor networks in underwater environments comes with associated challenges related to the communication layer. Because of the harsher conditions and related higher costs, but also because this environment comes with propagation scenarios that make radio-frequency communications difficult to use, the acoustic channel is preferred compared to radio communication [76,78], with typical data rates from 31 kbps to 125 kbps [88], or even up to 300 kbps in the solution by Freitag [79]. Optical communications are also a solution, and they provide faster, low energy communications with lower delays, but they require a line-of-sight unlike acoustic sensors. Some existing underwater sensor networks, such as S-Net in Japan, use wired optic communications. The most used communication choice for underwater wireless solutions is certainly the acoustic channel. Reviewed papers that used UWSN did not deal with ‘onland’ communications, which is usually carried out by an IP network such as a cellular network. Only [83] took into account the possibility of terrestrial communications failing in case of disaster, and as such considered the use of satellite communications to deal with that event. Satellite communication is also particularly suited to cover the large distances required in some solutions. Solutions such as LoRa and GSM have also been employed for tsunami warning devices.

Most TEWs based on a UWSN use a basic architecture which uses a central server connected to the sink nodes through an IP network. As such the main computing is done at said server, and there are different methods to detect the tsunami and its point of origin (inversion, Machine Learning, underwater sensors classification, etc.). Non-UWSN systems can also use simple threshold methods or other algorithms such as the Fuzzy Logic algorithm developed by [86]. These relatively simple systems also allow to move the prediction at the edge of the network and closer to the monitored area, making alerts faster.

### 6.3. Systems Requirements and Constraints for Tsunami EW Systems

An underwater acoustic network needs to be both reliable and energy-efficient. Accurate energy budgets/lifetime estimations are necessary since sensors replacement is more difficult, and energy harvesting techniques (wave motion harvesting, microbial fuel cells [89]) are suggested to enhance the life of the sensors deployed underwater [78]. The correct positioning of the sensors can help reduce costs and increase accuracy of predictions. The positioning of the central hub and terrestrial receivers is also important, since they could be compromised by earthquakes or tsunamis [85].

While the acoustic channel is a better solution for UWSN than the radio channel, it also comes with its own limitations: frequency dependent attenuation, multipath and high and variable delays. Specific modulation schemes are also suggested for the acoustic channel to enhance the overall communications and reduce multipath effects [89]. Routing protocols also play an essential role in UWSNs since acoustic communications are very energy consuming and therefore they require energy optimized routing, which has often been achieved at the expense of higher delays [78]. In IoUT, data aggregation is made more difficult by the fact that the overlap of multiple routing paths will increase packet collisions in the area and it will require re-transmissions, increasing energy consumption, and decreasing reliability [76]. When making use of mobile sensor nodes, such as underwater vehicles or sea gliders, further challenges in terms of routing, node localization and synchronization, and energy consumption are introduced [78].

Like for UWSN, buoys solutions for wave detection come with challenges compared to on-land deployments, since the connectivity between neighboring nodes can be affected by wave height, and as such dedicated routing protocols and processing algorithms are required [90], and the tilting of the buoys can also degrade the communications.

## 7. Landslides

Landslides are a recurring hazardous event consisting of the down slope movement of soil, rock, and organic material. There are different triggering factors to landslides, such as rainfall and changes in ground water levels, rapid snow melts, or earthquakes, and therefore different quantities can be monitored to detect and predict landslides, primarily displacement but also weather parameters in rainfall triggered landslides [18]. A Landslide EW system can be deployed at different scales (regional, national, or a more local scale), and IoT and MEMS based systems can be used to reduce costs and allow the installation of denser sensor networks. Rainfall is often a triggering factor for landslides, and as such it is useful to integrate available weather data or weather forecasts with the data obtained using the on-site sensors. Figure 5 illustrates an example implementation of a landslides Early Warning System for rainfall induced landslides.

### 7.1. Reviewed Uses Cases for Landslide EW

Table 7 summarizes the articles reviewed for Landslide Early Warning, while a more detailed review is included below.

Quoc et al. [91] developed and tested a WSN-based Early Warning System for landslides that uses a soil moisture sensor, accelerometers, and a Pore Water Pressure (PWP) sensor. To find a good trade-off between fault tolerance and battery saving, the sensors network uses a mixture of tree topology and star topology. Since a tree topology reduces energy consumption but it is more prone to failures because of clustering node failures (for example, during a landslide event), a star topology is also used in which nodes send data directly to a sink node. When the slope is not considered to be in a safe state then the network switches to a star topology; otherwise, the network uses a tree topology in which cluster nodes send data to a sink node placed in a safe location. The paper also takes into account synchronization and routing for the network. The data from the sensors is collected by a sink node and then sent to a central station for analysis, warning generation and decision making on the network topology, depending on the state of the landslide. The central station uses a thresholds-based system to determine a warning level using data from both a rain gauge station and the WSN.

It is not always easy to identify the right thresholds for a Landslide Early Warning System, even when historical data is available. Abraham et al. [92] developed a Landslide EW system in the Himalayas based on MEMS tilt sensors and water sensors embedded in the soil. They also studied the correlation between rainfall levels and on-site conditions to obtain better thresholds and avoid false alarms. The monitoring system they developed consisted of six low-cost units each equipped with antennas to send measurements to a data logger placed near the sensor nodes, with 10 min sleep intervals after each transmission. The tilt sensors were able to measure slow ground displacement, showing they are viable to predict failures. They also showed that rainfall is not always directly correlated with tilting variations and that both short-term and long-term rainfall should be considered to develop regional thresholds.

Gamperl et al. [93] developed a MEMS-based system that uses LoRa for communication. The lower layer of the Landslide EW system consists of: (i) a multi-sensors LoRa network with three types of sensor nodes and at least two LoRa Gateways serving each node for redundancy; and (ii) a Continuous Shear Monitor, piezometers, and extensometer systems, which require more space and are harder to install compared to the LoRa nodes. The LoRa nodes are therefore used to cover a large area and to monitor a wide array of environmental parameters at lower costs. A central station gathers data from the LoRa nodes and the other systems and forwards it to a central Cloud server called Inform@Risk. The Cloud manages and combines the data from the various sensors to produce Early Warnings and danger levels based on thresholds which are currently still being determined. Warnings are issued through an App and sirens installed locally, but alarms are immediately sent only when at least two neighboring nodes display strong accelerations at the same time and a data analysis algorithm has been run; otherwise, an expert is tasked to control the data to prevent false alarms.

Aggarwal et al. [94] proposed a landslide EW system that uses cameras to detect motion. They use a fairly simple motion detection algorithm so that it can be implemented on a Raspberry Pi connected to the cameras to detect landslides on-site. This way, the only transmissions are related to the images corresponding to a detected landslide event. The paper did not specify the preferred communication method. They also implemented a storage of events data on a Cloud database that can be accessed through a user application.

Thirugnanam et al. [21] developed a Landslide EW system that uses ML for predictions, an they also analyzed different factors to enhance the reliability of the system. They divide the system into five main components: (i) data collection, (ii) data transmission, (iii) forecasting, (iv) warning, and (v) response. By using Machine Learning algorithms in the system, they are able to obtain good results even if the first two layers fail, and they also stressed how not all current Landslide EW systems take into account improving the reliability of data transmission or adding redundancy. They consider two types of prediction: “nowcasting”, which uses rainfall forecast information instead of sensor data to still be able to produce a prediction when communications or sensors fail; forecasting, which provides extra lead-time for early warning in the fourth component by using data in the first two components when available.

Elmoulat et al. [95] proposed an Edge AI architecture for Landslide EW that can significantly reduce the latency of the system by doing the training of the ML model at the edge of the IoT network. The network consists of two types of LoRa nodes (Weather nodes and Ground nodes [98]) connected to a gateway, which communicates to an Edge AI cluster linked to a Cloud solution for storing and further processing using additional computing power. Compared to a Cloud-only approach, the developed solution allowed to reduce latency, bandwidth consumption, and amount of data transferred, even though the whole architecture had not been tested with a real event yet. A further improvement [99] was to introduce Federate Learning to the system, which is a distributed learning for Edge/Fog nodes which allows to continuously improve learning at this level and with different node resources.

Another Edge solution is the one by Joshi et al. [96], who proposed an architecture called Reliable LEWS. The goal of their architecture is to maintain the system working even if the connection to the Cloud server is interrupted by implementing Edge nodes that are able to run the ML model created and trained in the Cloud. While the Edge nodes in this solution are unable to train the ML models by themselves, they can still provide predictions when the connection to the Cloud is gone by using the previously stored trained model and the data from the sensor nodes, which communicate with the Edge nodes using Wi-Fi.

The study by Li et al. [97] developed a 5G Landslide EW system in China that monitors rainfall, ground fissure, and surface deformations in real time. Their solution uses two complementary modes for data transmission: (i) mesh mode, which enhances fault tolerance and reliability, and (ii) linear mode, which instead allows a more effective long range communication. The system also improves data reliability by using a dual communication mode with the BeiDou satellite module and GPRS for data transmission. The data from GNSS stations (used to track surface displacement), ground monitoring stations, and weather stations is continuously transmitted to a central server. They also developed a web interface that allows to visualize and monitor the data in real time. There are four levels for warnings which are chosen based on the tangent angle of the slope deformation.

### 7.2. IoT Architectures for Landslide EW Systems

Table 8 reports the main technologies used and developed for each layer of the reference IoT EW architecture.

As already mentioned, there are many different factors that can contribute to landslides, and as such different sensors can be used to detect them, and the right parameters to be monitored and sampling rates should be carefully chosen [91]. MEMS systems can be useful to reduce the cost of installations, and tilt sensors are particularly interesting because they do not need to be installed deep in the soil like inclinometers, and they require less expertise in installation than extensometers [92]. Since there could be the need to monitor wide areas, sensor placement should be evaluated carefully and based on previously performed risk analysis [93]. Rainfall is usually the main triggering factor for landslides [92], so rainfall historical data, if available, and its measurement through weather stations or radar systems should be integrated in such a system.

Some of the reviewed papers did not specify the communication protocols used, nor did they all plan for redundancy in case of data transmission failures, even though more than one paper underlined the importance of taking into account possible network outages and planned accordingly. Using new technologies such as 5G could improve data reliability, coverage, and speed.

Many Early Warning mechanisms for landslides are based on thresholds that can be obtained from both empirical or probabilistic models [18], and they need to be optimized to avoid raising false alarms. When it comes to Landslide EW systems, most solutions adopted a Cloud or central server based approach to data processing and did not process data directly at sensor or gateway level, but more recent articles such as [95,96] started to integrate Edge solutions into their Landslide EW system to reduce latency and also to provide better reliability in case the connection to the Cloud fails.

### 7.3. System Requirements and Constraints for Landslide EW Systems

The reviewed literature highlighted the following constraints and aspects that should be taken into consideration when developing a Landslide EW IoT solution: (i) importance of avoiding false alarms (ii) minimizing costs, for example using MEMS sensors, which also reduce the difficulty of installation; (iii) optimization of sensors placement through risk analysis, the choice of the right parameter to monitor and the right sampling rates for each sensor; (iv) limiting battery consumption; (v) enhancing resilience in case of extreme events, also for communications through redundancy or other methods [21,93]; and (vi) planning for failure for both the communication channel and data acquisition.

It is very important to avoid false alarms, for example optimizing thresholds or defining multiple thresholds based on different parameters [21,91]. Gamperl et al. also added the prospect of using data fusion techniques to reduce the effect of human interference on measurements. Some articles also considered asking experts to check the collected data if the warning level is not high enough to issue an immediate alarm.

Battery consumption can be limited both using simple sleep modes in between transmissions, which are the most energy consuming operation for a sensor node, or employing network topologies that allow to reduce the number of transmissions for each node.

Lastly, landslide systems can be damaged by the events they monitor and as such one should plan accordingly, for example placing sink nodes further from the monitored area, or using more than one gateway to serve each sensor, or planning for redundant communication, since a failure on the data acquisition or transmission levels could lead to a system unable to predict dangerous events [21].

## 8. Recommendations and Future Work

Table 9 lists some possible solutions and recommended options to satisfy the requirements that were previously defined for an IoT Early Warning System, based on the reviewed literature. It is to be noted that some of these suggestions still require careful planning before the final implementation, for example evaluating link budgets or received signal strength when using a specific communication solution, or measuring energy consumption.

While the requirements for each use case (floods, earthquakes, tsunamis, landslides) are widely different, for example in terms of latency, the table shows how Edge solutions can be used to satisfy different needs and as such should be taken into consideration during the development of an Early Warning system based on the IoT, regardless of the specific application. Edge Computing might also consist in the deployment of AI at the edge of the architecture. When using ML, learning at the Cloud might not be the best solution for wide monitored sites, geographically dispersed locations or rapidly changing environmental conditions [95], and as such Edge AI could be integrated into the architecture, keeping in mind possible additional processing time depending on the complexity of the ML model and the resources of the Edge devices. Besides ML, simpler algorithms or threshold based methods can also be employed on-site if the devices are limited in terms of computational capabilities. In conclusion, Cloud Computing should not be completely replaced by Edge Computing, but they should work in tandem to provide a better overall system.

Moreover, we give the following recommendations for future work and research, also taking into account some of the gaps or trends that were found in the reviewed literature:Optimization of MEMS sensors and widening of their employment. Since cost is an important factor in EW and IoT systems, they provide a valuable help in reducing costs while allowing to obtain dense sensor networks and the related benefits. MEMS sensors do not provide the same accuracy as other “standard” sensors, but they can still be used effectively in EW, and further research on sensors design and sensor signal conditioning can help improve their performance.Moving towards new cellular solutions such as 5G or NB-IoT, since some of the reviewed papers still relied on GSM or generally older cellular technologies. The use of new cellular solutions specifically designed to support IoT device and machine to machine communication (such as EC-GSM, NB-IoT or LTE-M) could provide better coverage, battery saving, data rates and latency performances in this context.Employment of non-terrestrial-networks and satellite communications. Satellites do not only allow to reach geographically distant locations and overcome the problem of missing terrestrial infrastructures [37], but they can also provide an essential back-end in case of disasters affecting the terrestrial communication infrastructures.Optimization of the prediction algorithms and data processing speed, especially for use cases such as earthquakes. Data analysis can become the bottleneck of an EW system and data latency is the main component of EEW latency [60], and as such efforts should be focused on its reduction for the use cases that particularly value speed. Besides the signal processing algorithms, packetization and data serialization formats should also be evaluated in terms of their impact on the system speed, and these aspects alongside others such as data logging management should become part of the system design.Employment of multi-parametric monitoring and modeling to enhance the accuracy of predictions, reduce false alarms and enhance the resilience of the system if some data collection modules experience faults. Rainwater forecasts, satellite ground motion data, satellite imagery, and terrestrial sensor networks provide data that can be used together to provide better forecasting models and thresholds optimizations. A semantic approach can help improve the performance of a system with a wide array of different data sources [11].

## 9. Discussion

The review showed that an IoT solution in the context of Early Warning can be very effective in the tasks of data collection, transmission, and disaster prediction, all the while retaining cost-effectiveness. As such, Wireless Sensor Networks, Cloud solutions, Machine Learning, and other components of the Internet of Things should be used when deploying Early Warning Systems, or integrated into already existing ones. This would allow a denser presence of alert systems on the territory and would therefore provide timely warnings and essential data and instruments to the authorities, assuring economical and societal benefits by reducing risks associated with disastrous events. Moreover, many of the requirements of an IoT-based Early Warning system can be completely or partly satisfied by adopting an Edge Computing solution, possibly alongside a Cloud one. This would bring the system resources closer to the end devices of the network with the aim of reducing latency, number of transmissions, and data processing at centrals servers. The review showed some already existing trends in this regard, with examples of Edge solutions, on-site predictions and Edge AI implementations, as well as other technological trends such as the more and more frequent adoption of low-cost MEMS systems and Machine Learning prediction algorithms.

The article also showed how an accurate choice of the network topology, or the implementation of mesh and adaptive solutions can be effective in satisfying other constraints such as coverage, reliability and battery management. The selection of the right communication protocol is also essential, both for the energy consumption and the coverage aspects. Ad hoc solutions such as LoRa or other Unlicensed protocols can be effective in areas where there is not good cellular coverage, but in this case the limits due to the shared radio resource need to be taken into account. Non-terrestrial-networks are also an important alternative to consider, since they could provide a more reliable back-end than terrestrial networks, which are more prone to failure in case of natural disasters, and can generally provide better coverage in areas that do not have good cellular infrastructures.

Some gaps in the recent literature were also found, particularly the lack of fault tolerant solutions, both in terms of communications and sensing node failures, and the use of non-state-of-the-art communication protocols, since, for example, some solutions still used older cellular technologies such as GSM.

The recommendations provided aim at filling these gaps and giving a meaningful contribution to the future research on EW systems, particularly regarding topics such as very low-latency communication, real-time monitoring and control of environmental parameters, very low-cost sensors, and high density sensor networks. Moreover, another important recommendation concerns the potential benefits of widespread sensor networks, since density can provide better fault tolerance but also improve the effectiveness of predictions and event detection.

## 10. Conclusions

This article reviewed papers that dealt with the proposal, development, testing, and optimization of Early Warning Systems for natural disasters that are based on the Internet of Things. In particular, four different use cases were taken into consideration: floods, earthquakes, tsunamis, and landslides. For all the four scenarios a reference three-layered architecture was used to better extrapolate the IoT solutions adopted in each paper, and to highlight the objectives, final results, and possible limitations of each work. Based on the reviewed literature, it was found that the use of the Fog/Edge computing in the developed architecture allows to reduce latency, number of transmissions, and data processing. Another finding was that in new IoT-based Early Warning systems more focus should be put into the fault tolerance capabilities of the deployed solutions, in terms of sensor networks and communications resilience. Moreover, some recommendations regarding battery consumption optimization, latency, communication efficiency and reliability should be taken into account for the improvement of existing systems or for future research.

## Figures and Tables

**Figure 1 sensors-22-02124-f001:**
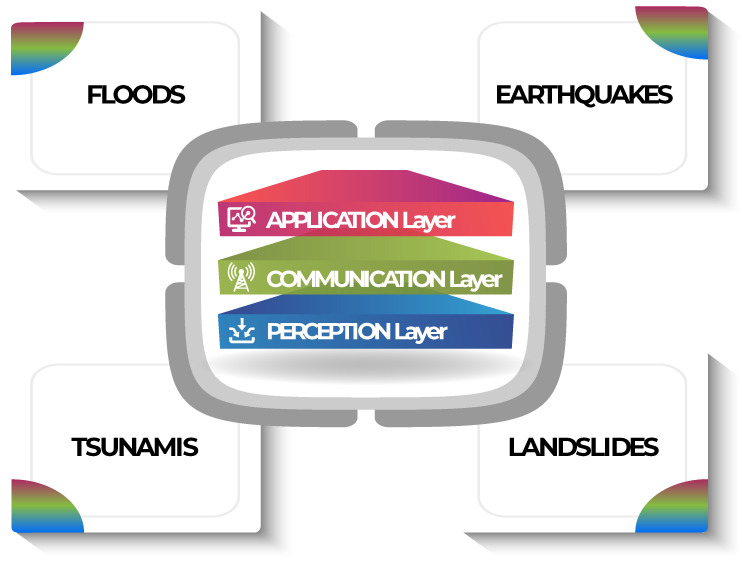
Reference IoT architecture.

**Figure 2 sensors-22-02124-f002:**
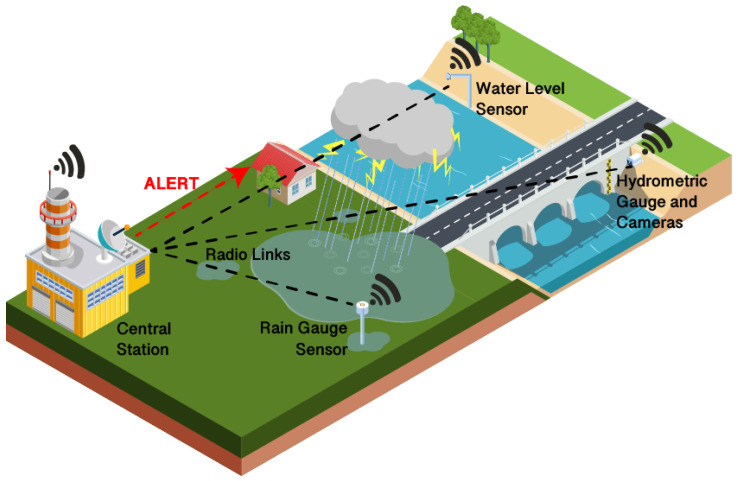
Example implementation of a Flood Early Warning System based on the IoT.

**Figure 3 sensors-22-02124-f003:**
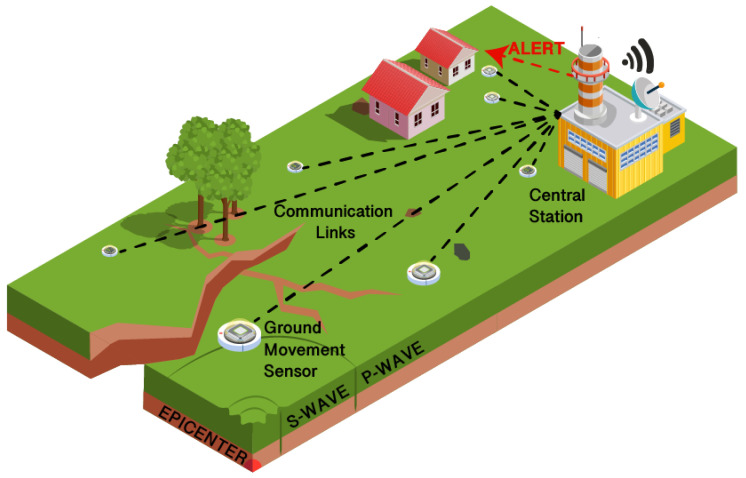
Example implementation of an Earthquake Early Warning System based on the IoT and P-wave detection.

**Figure 4 sensors-22-02124-f004:**
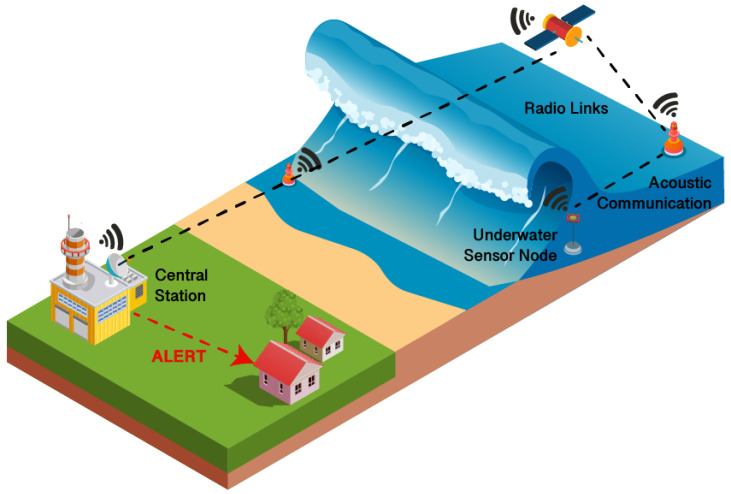
Example implementation of a Tsunami Early Warning System based on the IoT and underwater sensors.

**Figure 5 sensors-22-02124-f005:**
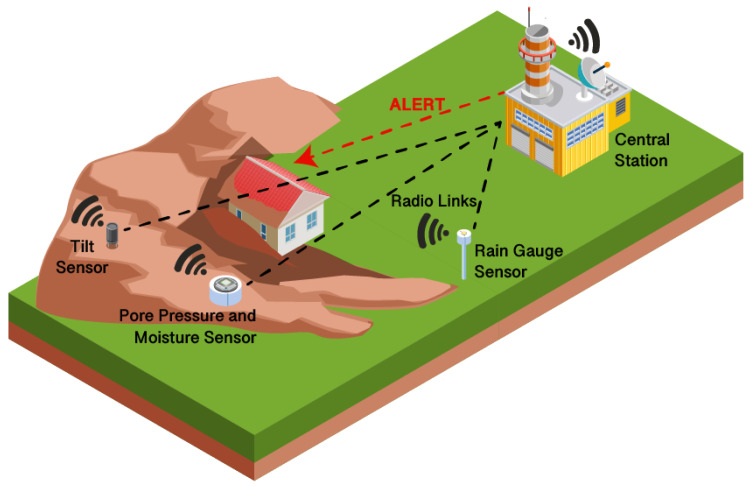
Example implementation of a Landslides Early Warning System based on the IoT and underwater sensors.

**Table 1 sensors-22-02124-t001:** Reviewed Articles in Flood EW.

Article	Focus and Objectives	Results
[46]	The paper proposes a simple IoT system for dams flood warning, trying to compensate some issues in previous solutions such as the need to deal with heterogeneous data, power consumption, and the abrupt loss of cellular coverage.	A cellular application is to be deployed to alert users, based on Zigbee tech. This would provide resilience to cellular network failures, but it requires users to use Zigbee hardware.
[26]	Fog Layer implementation in Flood EW to reduce latency and provide predictions before the Data Analysis Layer, and energy and cost optimizations through hexagonal division of the monitored area. Proposal and validation of prediction algorithms.	A prototype was proposed to test all layers simultaneously, showing effectiveness of the proposed architecture. Algorithms were tested with geographical data collected from flood monitoring websites.
[47]	Use of different sensors connected using 3G/2G/GSM modules and LoRa modules, taking advantage of the good network coverage in the area, to process, compile, and forward information to provide timely warnings.	The cost of this solution is high, but it proved successful in generating alarms and it comes with an user interface.
[48]	Revival of a rural water monitoring system, with a focus on how NLoS communication and remote locations are challenging for GSM, and how an ad hoc LoRA/non-cellular network could be an attractive and reliable option.	Experimental measurements on the LoRa transmission show a dependency on the receiver and transmitter heights and worse performance than empirical models. A migration to other cellular protocols besides GSM is suggested.
[49]	The study develops a river monitoring system with a focus on power consumption, showing that radio transmissions are the most energy consuming operation for the node and how critical measurement modes affect energy waste.	When solar batteries are not recharged and when in critical mode, battery life of the sensor node becomes 57% shorter. The measurement success rate also depends on the cone used for the ultasonic sensor.
[50]	Sensor node implementation for Flood EW WSN with low energy consumption achieved through a dedicated deep sleep algorithm and testing to evaluate power consumption	Current is on average 13 mA during the setup time period, and it drops to an average of 0.3 mA in standby phase. Thanks to short wake-up and transmission windows, more than 280 days are needed to discharge a 2200 mAh battery. A full WSN implementation and high traffic were not tested yet.
[44]	Case study for an EW system called SENDI with a focus on fault tolerance through a clustering model and sensor nodes with embedded forecasting models and self-organizing algorithms.	The clustering system and the forecasting models were tested through simulation, showing good results as far as energy optimization, fault tolerance and accuracy. A limitation is the requirement of a large number of nodes to properly work.
[51]	Development of a WSN whose data is used to train and validate a SVM Machine Learning model which can be embedded into the WSN coordinator nodes.	98% accuracy for flood prediction was achieved. Architecture thresholds are dependent on the location, so the system needs to first obtain data (if not available) and compute them. There is ambiguity for data from overlapping endpoints. LoRa communication worked with virtually no data loss.
[52]	Development of an Early Warning system based on Computer Vision, using images from a CMOS cameras sensor network and comparing them with stored references using image comparison techniques.	The developed algorithms are able to provide curves regarding: (i) rank of calamity, (ii) image registration curve, (iii) precision, (iv) object recognition curve. A web platform was also developed to make the graphs available to users.
[53]	Proposal of a cloud based wireless sensor and actuators network to alert from floods and to open watchman inlets to make water flow out of the monitored water source if needed.	An algorithm to manage the sensors and actuator was also and successfully validated through model analysis, specifying different checks for water. The paper did not propose a specific communication protocol to send data over the Cloud or to the actuators.

**Table 2 sensors-22-02124-t002:** Technologies and solutions used for each layer of the reviewed papers for Flood EW.

Article	Perception Layer	Communication Layer	Application and Edge Layer	Edge/Fog Computing
[46]	Water Level, Water flow	Zigbee	Data analysis and threshold-based warning at remote server; Android smartphone application	No.
[26]	Water flow sensor, water level sensor and rain value.	None proposed	Holt–Winter method for prediction at remote server; early processing and prediction at sensors/gateway level	Yes.
[47]	Monitoring, meteorological, and mobile stations	3G/GSM, LoRA	IoT Cloud Platform to receive, manage, and store sensor data; semaphore alert system	No.
[48]	HMS stations	GSM, LoRA	Data processing and storage at remote server	No.
[49]	Ultrasonic Sensor	EnOcean	Water level monitoring at remote server	No.
[50]	Resistive Sensors	LoRa	Threshold-based alarm system at Cloud level; data storage and analysis on separate website	No.
[44]	Water pressure, rain gauge	6LoWPAN	Machine Learning forecasting at Cloud and sensor level; Visual and sound alarms system	Yes.
[51]	Wind sensor, humidity sensor, temperature sensor and water level sensor	LoRa	SVM model embedded in concentrator nodes for risk classification; final decision at Cloud server	Yes.
[52]	CMOS Cameras	Zigbee, GSM	Web platform; algorithms to get information such as rank of calamity and precision	No.
[53]	Water Level, Water Volume	None proposed	Threshold-based alert system; Cloud solution to manage actuators	Yes.

**Table 3 sensors-22-02124-t003:** Reviewed Articles in Earthquake EW.

Article	Focus and Objectives	Results
[53]	Proposal and testing of an SWEDA earthquake algorithm to achieve milliseconds earthquake detection and warning and optimizing computational costs, also designing and producing 2 kinds of sensor nodes. The system is also able to detect different kinds of seismic waves.	The solution was tested with three different systems installed in SHM sites at Qatar University, and it identified p-waves 11 s before a second trigger for s-waves was generated, showing good Earthquake EW capabilities.
[62]	Proposal of a distributed Edge Computing approach for EEW which uses GPS stations and seismoters to run a ML model. The distributed approach would reduce the amount of data to be transmitted towards the Cloud and provide a more resilient system in case of wide earthquakes.	GPS stations and seismometers have a complementary performance (for large earthquakes the prediction accuracy is 99% for GPS, 28% for seismometers). As such, they evaluated a multi-parametric model, confirming how a combined model is the better choice in terms of accuracy.
[63]	Modelling and simulation of a WSN for Earthquake EW and evaluation of how its performance changes depending on different WSN parameters, such as synchronization protocol, sampling frequency and node density.	TPSN gives velocity and localization errors 0.6 to 6 times and 1 to 5 times greater than with perfect synchronization, but errors are 5 to 266 times and 96 to 361 times greater with no synchronization. Using 5 nodes, a faulty node corrupts 25% of the data, while with 25 nodes only 4% of the data is corrupted. A higher sampling frequency also improves performance.
[64]	Development and testing of a sensor node for dense Earthquake EW networks, achieving low costs using MEMS sensors and standalone TCP/IP units to handle communications, so that the device can be controlled by a simple and cheap MCU.	Data from the low cost sensor arrays have good consistency with data from standard seismographs, and seismic phases obtained were accurate enough for EW. Even small earthquakes were identified within a 20 km range from the sensors. Bigger earthquakes (M 4.7) were detected at up to 200 km from the sensors.
[65]	Performance Evaluation of a Dense MEMS-Based Seismic Sensor Array with 170 sensors (based on the design developed in [66]) to show that they can be classified as class B sensors.	The proposed sensors duplicate the performance of ‘traditional’ earthquake sensors with a percentage difference below 10%, also when far from the epicenter (150 km). Stations with difference higher than 20% were due to calibration errors.
[67]	Design and testing of a low-cost, low-power cloud based Earthquake EW platform called Earthcloud, which uses geophones to detect strong motions and issue first alarms to AWS IoT using MQTT, and then further processes the data using Amazon Kinesis for precise alarms.	One out of the three sensors used in the testing of the Earthcloud solution failed (it could not differentiate background noise and earthquake data), while the other two showed results which were comparable to the ones obtained by the national authority.
[68]	The articles proposes and validates a 3-layered architecture that uses smartphones to create a MCS widespread Earthquake EW network, using a layer intermediate to the sensors and the application layer to ensure reliability.	Validation shows a 12 s anticipation of the seismic peak, but the validation phase is still to be improved. Low power usage (4.9%) was achieved.
[69]	Overview and valuation of the performance of the MyShake smartphone earthquake monitoring application through simulation, with a focus on the possibility to use it for EW.	MyShake phones detect the earthquake 3.8 s after the origin time and the system recognizes the event when 26 phones have been triggered, 15.6 s before the S-wave reaches the monitored city, but slightly underestimating the magnitude. Alert delivery latency was of 2.8 s, but it could be reduced and it is unclear how it would scale with many devices.
[35]	Proposal of an Early Warning system that uses SHM to detect seismic events and evaluation of the advantages of a 5G gateway in this context, using the URLLC service.	Latency will be reduced (<1 ms) while maintaining very high reliability using URLLC. While the proposed system architecture and 5G networks provide advantages in two key Earthquake EW constraints, field tests still need to be carried out.
[70]	Performance evaluation of an energy economical WSN for SHM embedded with an Earthquake EW system, employing WoR modules to minimize the energy consumption required to wake up the WSN when an earthquake is detected with a P-wave detector.	WoR modules and radio triggering allow the nodes to go into complete sleep mode when no sensing is required, reducing power consumption to 350 μA with a 229 ms delay for waking up the nodes, with a better performance IEEE 802.15.4.

**Table 4 sensors-22-02124-t004:** Technologies and solutions used for each layer in the reviewed papers related to Earthquake EW.

Article	Perception Layer	Communication Layer	Application and Edge Layer	Edge/Fog Computing
[53]	Inclinometers	CANopen	SWEDA Algorithm (on-site P-Wave detection)	Yes.
[62]	GPS Stations, seismometers	Not Specified	Machine Learning classifiers both at sensor and central server level	Yes.
[63]	Seismometers	Model based on TDMA	Crosscorrelation-based multi-station algorithm to locate epicenters	No.
[64]	MEMS Accelerometers	Ethernet, 3G/4G	P-Wave detection algorithm and processing at remote server	No.
[65]	MEMS Accelerometers	3G/4G	P-Wave detection algorithm and processing at remote server; real-time Shake Maps computation	No.
[67]	Geophones	Ethernet	Cloud Based solution based on AWS IoT for data storage processing and P-Wave detection	No.
[68]	Smartphone MEMS sensors, GPS modules.	Wi-Fi, 2G/3G/4G	Smartphone application with classification algorithm; Machine Learning prediction algorithm at central server	Yes.
[69]	Smartphone MEMS sensors	Wi-Fi, 3G/4G	Smartphone application with ANN-based detection algorithm; cluster-based EW algorithm at central server	Yes.
[35]	Optic fibre, MEMS and piezoelectric sensors.	MBUS radio 169 MHz, 5G	Threshold-based sentinel node; SHM-based event monitoring	Yes.
[70]	SHM sensors, P-Wave Detector.	IEEE 802.15.4, 3G/Wi-Fi	P-Wave Detector on-site node; SHM-based event monitoring	Yes.

**Table 5 sensors-22-02124-t005:** Reviewed Articles in Tsunami EW.

Article	Focus and Objectives	Results
[79]	The study focuses on characterizing an underwater communication link for Tsunami Early Warning	The tested link showed high reliability with a 350 bps data rate, while higher rates, even if still achievable, provide a less reliable connection. The 1 s latency for the receiver alongside the good data rates makes this solution feasible for real-time tsunami warning.
[80]	Method to determine an optimal array configuration of offshore sensors for near-field Tsunami EW, also in terms of deployment costs.	The study was able to prove that a 3-sensor configuration is able to provide accurate estimations. It also highlighted some factors that influence predictions, such as depth, time of arrival and position of the sensors.
[81]	Method for near-real time tsunami forecasts using the S-Net network. It classifies sensors based on their distance from the uplift region and then computes the region’s area.	The technique was successfully validated on two previous tsunamigenic events, and it required 1 min to run the classification algorithm and not more than 10 min to estimate the source area of the tsunami.
[82]	Proposal of a framework for a WSN to produce Tsunami EW system detecting magnetic field, animal behavior and tide behavior changes and using an ML model.	The tested ML model showed that the most interesting features in terms of prediction were tide level changes and migration pattern changes.
[83]	Proposal of an EW system to improve the performance of existing Tsunami EW systems, particularly in terms of cost and compromised communications.	The produced vehicles have low mobility and a mostly passive operational mode, allowing low power consumption. The use of satellite communications and offline back-ends would allow the system to survive communications fails in case of disaster.
[84]	Design of a device for wave height monitoring using an ultrasonic sensor placed above water, directly on the shore line. When the water level rises, a GSM module also sends an SMS with a warning message.	The prototype was successfully tested but the sensor cannot work at heights above 3.19 from seawater, and as such it requires a careful installation.
[85]	Development and deployment of a buoys system that uses buoys positioning measurements to monitor the motion of sea surface, also sending other data such as sea level. An acoustic system was also deployed to monitor the ocean’s bottom motion.	Field experiments showed centimeter precision in buoys positioning. An experiment showed data gaps, possibly due to the tilting of the buoy degrading the communication, which led to the implementation of a system to stabilize the antenna.
[86]	Design of a floating IoT device for early warning and wave anomalies detection, using Fuzzy Logic for accurate prediction.	The Fuzzy Logic algorithm had high accuracy (98% to 100%) in testing phase. The LoRa communication showed 4.6279 s of delay and minimal error rate.

**Table 6 sensors-22-02124-t006:** Technologies and solutions used for each layer by the reviewed papers for Tsunami EW.

Article	Perception Layer	Communication Layer	Application and Edge Layer	Edge/Fog Computing
[79]	Pressure Sensors.	Acoustic communications.	Not yet implemented, the goal is to use the optimized network for real-time detection.	No.
[80]	Methodology independent of the type of sensor	Not specified	Tsunami forecasting based on "Inversion for initial sea-Surface Height" method to determine tsunami source	No.
[81]	Optic Communications	Pressure Sensors.	Classification of sensor nodes at central server	No.
[82]	Magnetic sensors, tide gauge, count sensors and motion sensors	GSM	Machine Learning model at central server	No.
[83]	Autonomous underwater vehicles	Acoustic signals processing; Satellite communications	Warning support decision platform able to work offline	No
[84]	Ultrasonic Sensor	GSM	Threshold-based SMS alarm system at sensor level	Yes.
[85]	Satellite Communications	GNSS system, acoustic sensors	Coordinates estimation at buoy level; buoy data processing and dissemination at central station; web server user interface	No.
[86]	Gyroscope, accelerometer	LoRa	Fuzzy logic algorithm at sensor level; data processing and storage at central server	Yes.

**Table 7 sensors-22-02124-t007:** Reviewed Articles in Landslide EW.

Article	Focus and Objectives	Results
[91]	Development and testing of a threshold-based Landslide EW system that uses a mix of tree topology and star topology to achieve a good trade-off between energy consumption and reliability.	The system was implemented and it showed better power saving than previous work thanks to the mixed network topology and to the possibility to switch the sensors between a warning mode and a normal mode. LoRa is suggested for long range communication instead of Zigbee if needed. The system requires a geotechnical investigation to determine thresholds.
[92]	Development of a landslides warning system that uses MEMS tilt sensors and water sensors with a focus on avoiding false alarms through on-site parameters.	The system was tested and the tilt sensors were successful in monitoring slow ground monitoring for EW; it was observed that tilt variations are not always related to rainfall levels and that local site conditions should be considered besides thresholds based on rainfall.
[93]	Design of low-cost LoRa sensor nodes to be used alongside more classic ground monitoring sensors. A central station gathers data from the LoRa network and the other sensors and sends it to a Cloud solution called Inform@Risk which processes data to generate alarms.	The low cost sensors that were developed can allow a wider and more flexible sensor network. A problem of the system is that it might not be able to detect small landslides or issue warnings if there are not significant surface deformation, as it is a shallow system. A larger installation of 130 sensors will soon be tested in Colombia.
[94]	Camera-based Landslide EW system that uses a simple image processing algorithm that can be implemented on-site to limit images transmission.	The developed system was able to send a warning to the developed application 5–6 s after the event was detected to alert users.
[21]	Development of a Machine Learning-based warning system and analyses of Landslide EW architecture to improve their performance and reliability. The predictions are obtained from two algorithms: a nowcasting algorithm and a forecasting one.	The nowcasting algorithm had a 95% accuracy in a real scenario and it was a valid solution in case of data acquisition or transmission failures. The forecasting algorithm also implemented allows extra early warning time when the sensor data is available.
[95]	Proposal of an Edge AI architecture to limit transmissions towards the Cloud and the amount of data to be transferred on the network through processing of sensor data in a local Edge cluster	Network latency was reduced from 208 ms to 53 ms, and bandwidth consumption from 2.63 Mbps to 249.5 kbps, but data processing lasted up to 12 s.
[96]	Proposal of an Edge computing early warning solution which allows Edge nodes to run ML models on-site.	The system increases reliability by making the Edge Nodes predictions independent from the Cloud, but the developed Edge nodes are unable to train models by themselves.
[97]	Development of 5G Landslide EW system that monitors rainfall, ground fissure, and surface deformation in real time, and generates warnings based on slope deformation, and also provides other communication channels for reliability.	The implemented communication system allows transmission up to 100 km and it includes redundant channels. The system is able to produce both user interfaces and warnings.

**Table 8 sensors-22-02124-t008:** Technologies and solutions used for each layer of the reviewed papers for Landslide EW.

Article	Perception Layer	Communication Layer	Application and Edge Layer	Edge/Fog Computing
[91]	PWP sensors, accelerometers, soil moisture sensor, rain gauge station	Zigbee, GSM/GPRS	Threshold-based warnings at central station	No.
[92]	Tilt Sensors, volumetric water content sensors	Not specified wireless communication	Data aggregation into local datalogger; processing and analysis at remote server	No.
[93]	Continuous Shear Monitor, piezometers and extensometer, inclination sensor, water sensor.	LoRa, GSM	Data analysis and threshold-based warnings at Cloud Server; alarms through Smartphone Application and local sound systems	No.
[94]	Camera Sensor	Not specified	Landslide detection performed locally; storage of images to database; alarms through Smartphone Application and SMSs	Yes.
[21]	Rainfall, pore water pressure sensors	Not specified	Data analysis at central server; forecasting and nowcasting ML models.	No.
[95]	Weather nodes (temperature, humidity pressure, rainfall, wind speed), ground nodes (GPS, soil moisture, accelerometer, gyroscope)	LoRa	Edge AI implementation for prediction and monitoring; Cloud server for storage and additional processing power	Yes.
[96]	Moisture, porepressure and displacement sensors	Wi-Fi, GSM	ML model training at Cloud server; ML model prediction at Edge node.	Yes.
[97]	GNSS station; ground crack monitoring station; rainfall monitoring station.	5G, GPRS	Web platform to analyze data and issue warnings based on slope deformation	No.

**Table 9 sensors-22-02124-t009:** Recommended solutions for each of the main EW constraints.

Requirement	Recommendations and Possible Solutions
Battery Saving	Employing smart sleep procedures in-between transmissions, keeping short wake-up and transmission windows [50,70], or dynamically switching to sleep mode based on the event probability of occurrence in the area [26]. Choosing the right network topology or employing multiple topologies in the same solution [91]. Using energy harvesting modules [49]. Distinguishing between a "normal" mode of operation and a "danger" mode with higher sampling frequencies [49,91]. Using Edge implementations and on-site warnings to limit the amount of transmissions towards higher layers of the EW architecture [62,95].
Fault Tolerance	Using mesh topologies and self-repairing/self-organizing networks [44,54]. Planning for redundancy in communication [93] or for non-terrestrial satellite back-ends or communication links [83]. Employing a higher nodes density, also to limit prediction corruption from faulty nodes [63]. Implementing "nowcasting" procedures, if possible for that use case [21]. Using Edge implementations to be able to issue warnings even if the connection to the Cloud is down [96] or to distribute the data processing at different locations [62].
Coverage	Using ad hoc networks when cellular infrastructures or communications are inefficient [46,48]. Making use of satellite communications to reach difficult locations that are not served by cellular communication [37].
Latency	Integrating 5G technologies into the project, particularly its low latency services [35]. Using on-site methods and Edge implementations to be able to limit response time in time critical applications [39].

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
