# Peer review of "Recent Advances in Internet of Things Solutions for Early Warning Systems: A Review"

_sensors, 2022, doi:10.3390/s22062124_

Round 1

Reviewer 1 Report

The manuscript presents the “Recent Advances in Internet of Things Solutions for Early

Warning Systems: a Review”. Several suggestions would immediately improve the paper's readability and are severely lacking in the current manuscript.

  1. A lot of grammatical mistakes. This manuscript should be proofread.
  2. The organization of the paper is missing. Add it at the end of the Introduction section.
  3. What is the author's contribution? Highlight it.
  4. Recent studies are not included in this paper, such as

[1] Wanare, Ram, Kannan KR Iyer, and Prathyusha Jayanthi. "Recent Advances in Early Warning Systems for Landslide Forecasting." Geohazard Mitigation (2022): 249-260.

[2] Qi, Li, Zetian Wang, Di Zhang, and Yunfa Li. "A Security Transmission and Early Warning Mechanism for Intelligent Sensing Information in Internet of Things." Journal of Sensors, (2022).

[3] Rangra, Abhilasha, and Vivek Kumar Sehgal. "Natural disasters management using social internet of things." Multimedia Tools and Applications (2022): 1-15.

I suggest the authors include the recent studies from 2021/22.

  1. Add a discussion paragraph before the conclusion and discuss the findings.
  2. The conclusion is so long. Short it.

Reviewer 2 Report

This paper reviewed the literature about early warning system by emphasizing the IoT solution parts. Overall, the review has provided meaning summarization and provide some insights for future research direction. However, several comments below can be addressed before its suitability for publication: - In the paragraph of Introduction, it is beneficial if the authors provide the reason why early warning system (EWS) is important by emphasizing its (expected) benefits or impacts if EWS is used or applied. Some statistics would be useful to indicate its important. - In introduction it is also needed to mention related paper who also reviewed the IoT for EWS and compared with the current aim of this study. Emphasize the unique contribution of the present study. - Some figures are needed to illustrate the flow of this review papers. - How many papers have been collected, filtered, and selected? - To give better understanding for the reader, please provide the illustrations/figures for each of type of disasters: floods, earthquakes, tsunamis and landslides on how the IoT is used in EWS.

Round 2

Reviewer 1 Report

The authors of this manuscript have been revised according to my comments. Now, It can be accepted in its present form.

Reviewer 2 Report

All comments have been nicely addressed and there’s no more comment about this paper.